# Pruning Long Chain-of-Thought of Large Reasoning Models via Small-Scale Preference Optimization

**Bin Hong**[1]**, Jiayu Liu**[1]**, Kai Zhang**[1]**, Jianwen Sun**[3]**, Mengdi Zhang**[4,5]**, Zhenya Huang**[1,2,†]

[1] State Key Laboratory of Cognitive Intelligence, University of Science and Technology of China
[2] Institute of Artificial Intelligence, Hefei Comprehensive National Science Center
[3] Artificial Intelligence in Education, Central China Normal University
[4] Meituan      [5] NeoShell
hb2002@mail.ustc.edu.cn mdzhangmd@gmail.com huangzhy@ustc.edu.cn

## Abstract

Recent advances in Large Reasoning Models (LRMs) have demonstrated strong performance on complex tasks through long Chain-of-Thought (CoT) reasoning. However, their lengthy outputs increase computational costs and may lead to overthinking, raising challenges in balancing reasoning effectiveness and efficiency. Current solutions often compromise reasoning quality or require extensive resources. In this paper, we investigate how to reduce the generation length of LRMs with limited tuning. We analyze generation path distributions and filter generated trajectories through difficulty estimation. Subsequently, we analyze the convergence characteristics of various preference optimization objectives under a unified Bradley-Terry loss based framework. Based on the analysis, we propose Length Controlled Preference Optimization (LCPO) that directly balances the implicit reward related to NLL loss. LCPO can effectively learn length preference with limited data and training. Extensive experiments demonstrate that our method significantly reduces the average output length of LRMs by over 50% across multiple benchmarks while maintaining the reasoning performance. Our work highlights the potential for computationally efficient approaches in guiding LRMs toward efficient reasoning.

## 1 Introduction

Building upon Large Language Models (LLMs), numerous powerful models have emerged, significantly enhancing AI capabilities across diverse domains (Jiang et al., 2025; Guo et al., 2024; Wang et al., 2024). Recently reasoning-oriented LLMs, or Large Reasoning Models (LRMs) such as Deepseek-R1 (Guo et al., 2025) and QwQ-32B (Team, 2025; Yang et al., 2024b) have achieved remarkable performance on complex reasoning tasks. These models learn to perform long Chain-of-Thought (CoT) (Wei et al., 2022b) thinking through online reinforcement learning with verifiable reward (RLVR) (e.g., PPO (Ouyang et al., 2022) and GRPO (Shao et al., 2024) with conventional accuracy reward), which explicitly decomposes the given problem and proceeds to solve each subproblem in a step-by-step manner. Furthermore, they actively recall and verify previously generated steps throughout the reasoning process (Cheng et al., 2025; Xue et al., 2024). This significantly increases the output length while empowering LLMs to effectively address reasoning-intensive tasks.

The tendency of LRMs to generate excessively long responses introduces additional challenges (Feng et al., 2025; Qu et al., 2025; Liu et al., 2025b; Xu et al., 2025c; Liao et al.). As shown in Figure 1, for an easy question, an LRM can consume 5465 tokens to solve it. On one hand, this tendency significantly increases computational and memory costs when using LRMs for various downstream tasks, which limits the applicability of continual learning for further enhancing the capabilities of LRMs. On the other hand, overly long outputs may lead to overthinking, where LRMs consume more tokens than standard models and may produce redundant or even incorrect

---

[†]Corresponding author

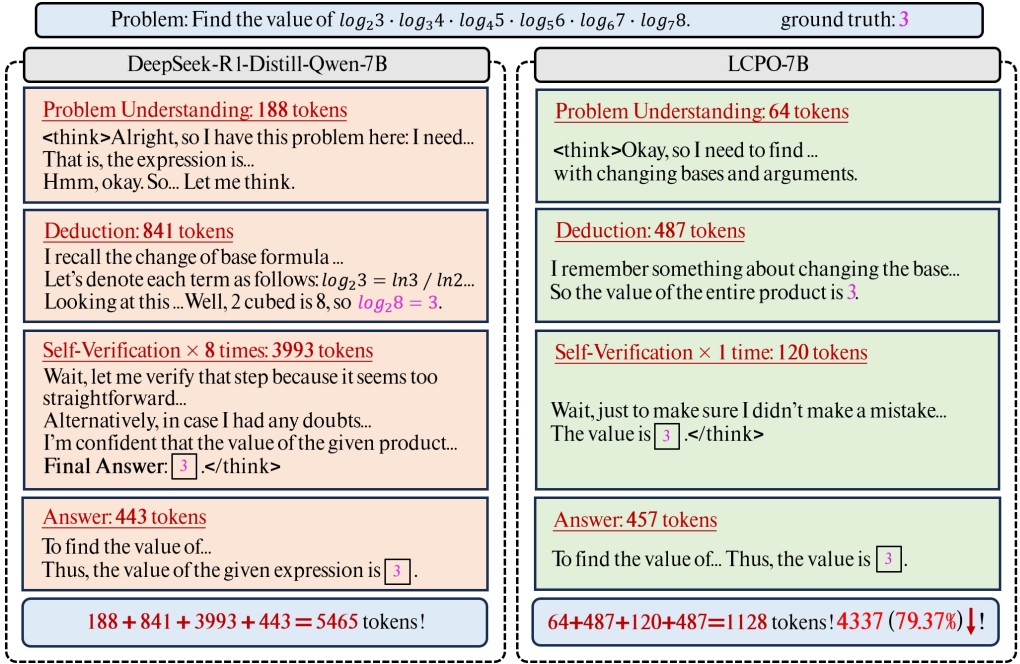

Figure 1: An example math problem from MATH-500. Left: the lengthy output of LRM. Right: the concise output of LRM after trained with LCPO.

outputs on relatively simple queries (Sui et al., 2025). These challenges have prompted a renewed focus on a critical question for LRMs: *How to enable efficient reasoning with shorter output length and maintained reasoning performance?*

Exploration of this question remains limited. Inference-time pruning methods (Xu et al., 2025b; Muennighoff et al., 2025) append length-related tokens to the prompt or generation process to encourage earlier termination. While these approaches are low-cost, they are unstable in reducing length, and often compromise the model's reasoning capabilities. Subsequent works (Aggarwal & Welleck, 2025; Hou et al., 2025; Arora & Zanette, 2025) adopt a large-scale multi-objective RL paradigm. They introduce a length-regularized reward alongside the conventional accuracy reward and perform RL on a large amount of data, aiming to reduce generation length and recover reasoning capacity. Online RL approaches involve complex training systems and higher resource demand compared to offline fine-tuning (Yu et al., 2025; Chen et al.), which contradicts the idea of efficient reasoning. Additionally, when incorporating budget-forcing mechanisms where a token limit is set in the prompt (Muennighoff et al., 2025; Aggarwal & Welleck, 2025), online RL-based methods are inherently constrained by manually predefined token budgets, limiting their adaptability to dynamic task requirements and can again compromise reasoning capabilities.

To address these limitations, we investigate how to reduce the generation length of LRMs with limited tuning. As RLVR biases the output distribution toward rewarded trajectories (Yue et al., 2025), we explore biasing the trajectory distribution explicitly within the LRMs generation space (Yue et al., 2025) in an offline manner. In this paper, we ask the following research questions: RQ1: Is there still shorter yet effective reasoning paths within the generation space of reasoning models? RQ2: How can we leverage limited training and data to adjust reasoning model's generation distribution?

To answer these questions, we empirically analyze the trajectory distribution of generation space of LRMs. We use the pass rate as a proxy for the perceived difficulty of a problem and leverage this metric to reduce data volume while reserving concise generation patterns. Subsequently, we employ preference optimization as an efficient self-distillation training method to push the output distribution toward the shortest effective paths. To find an ideal algorithm for effective small-scale training, we theoretically analyze the convergence conditions of different objective functions used in various preference optimization methods under a Bradley-Terry loss framework. As discussed in Appendix D, we find the implicit reward associated with Negative Log Likelihood (NLL) loss can

Table 1: Data needs for rollout and training of different efficient reasoning methods.

| Method | Type | Data | Train Samples |
|---|---|---|---|
| CoD (Xu et al., 2025b) | inference-time | - | - |
| L1 (Aggarwal & Welleck, 2025) | RL & budget-forcing | 645k | 645k |
| TrEff (Arora & Zanette, 2025) | RL | 24.8k | 24.8k |
| DAST (Shen et al., 2025) | preference optimization | 150k | 20.6k |
| Ours | preference optimization | 22k | 0.8k |

hinder length preference alignment. Based on this analysis, we propose Length Controlled Preference Optimization (LCPO), which incorporates an explicit term of identical form to counterbalance the implicit NLL-related reward. This enables precise length alignment with small-scale training data, yielding an efficient, low-cost, and computationally lightweight reasoning pipeline.

We conducted extensive experiments using DeepSeek-R1-Distill-Qwen-1.5B and DeepSeek-R1-Distill-Qwen-7B, following previous works (Aggarwal & Welleck, 2025; Arora & Zanette, 2025) on many widely used benchmarks. Our method reduces the average generation length by over 50% across all benchmarks while maintaining the original model's reasoning performance. As illustrated in Figure 1, our approach can prune unnecessary parts of the reasoning trajectory, reducing length by 79.37%, and guides the model to more concise reasoning. Furthermore, as shown in Table 1, our method requires only 0.8k training samples and 50 training steps, largely reducing computational cost compared to prior approaches. We believe our findings and method provide valuable insights for future research on efficient reasoning in LLMs.

In summary, our main contributions are as follows:

- We investigate the generation space of LRMs and introduce a simple yet effective data filtering strategy that preserves a more concise generation mode. Under a unified Bradley-Terry loss framework, we further analyze the convergence characteristics of different preference optimization methods, providing insight of offline alignment objectives.

- We propose Length Controlled Preference Optimization (LCPO), an objective specifically designed to effectively learn length-related preferences from limited training data, offering enhanced efficiency and generalization ability compared to existing techniques.

- We conduct extensive experiments on several popular reasoning benchmarks. The results demonstrate that our method significantly reduces average output length by over 50% across different benchmarks while maintaining reasoning performance. We also perform further experiments to gain deeper insights into the characteristics of our method.

## 2 RELATED WORKS

**Reinforcement Learning on LLM.** Recent works typically employ reinforcement learning with verifiable reward (RLVR) to build models with enhanced reasoning capabilities (Guo et al., 2025; Jaech et al., 2024; OpenAI, 2025) through online RL algorithms such as Proximal Policy Optimization (Schulman et al., 2017; Ouyang et al., 2022) (PPO) and Group Relative Policy Optimization (Shao et al., 2024) (GRPO). However, this paradigm involves alternating generation with parameter updates, which increases training complexity. It typically demands more computational resources and is significantly slower compared to offline tuning methods. Direct Preference Optimization (Rafailov et al., 2023) (DPO) is an offline alternative to online RL algorithms, introducing implicit reward via pairwise preference data through the Bradley-Terry model. Subsequent advancements including SimPO (Meng et al., 2024), SimPER (Xiao et al., 2025) and ORPO (Hong et al., 2024) further eliminate the need for the reference model in the DPO objective.

**Test-Time Scaling and Efficient Reasoning.** LRMs often generate excessively lengthy outputs to enhance reasoning abilities (Chen et al., 2025; Xu et al., 2025a), known as Test-Time Scaling (Wei et al., 2022a; Wang et al., 2022; Zhang et al., 2025b; Liu et al., 2025b; Feng et al., 2025). To reduce

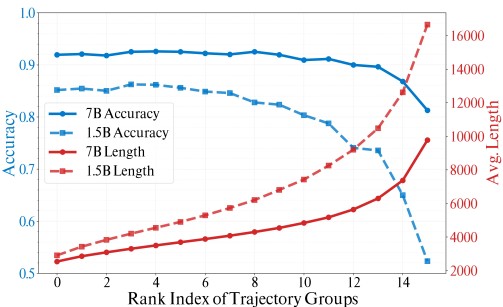 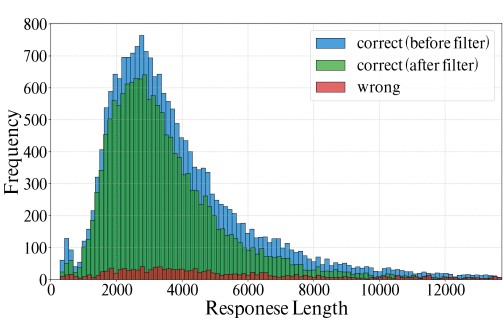

(a) Performance of 7B and 1.5B models on LIMR with varying average output lengths.

(b) Distribution of output length from 7B models on LIMR before and after filtering

Figure 2: (a) As the number of output tokens increases, model performance tends to deteriorate. The x-axis represents the sorted rank of the samples. (b) After filtering, the distribution change is minor.

the computational overhead and redundancy introduced by this, several studies (Muennighoff et al., 2025; Zeng et al., 2025; Xu et al., 2025b; Arora & Zanette, 2025) incorporate length-related tokens or impose explicit constraints in prompt or generation process, which often compromise models' performance. Besides, explicit length penalties are naturally leveraged in the reward computation of RLVR for shortening reasoning trajectories (Team et al., 2025b; Aggarwal & Welleck, 2025). This approach typically requires large-scale training with high resource demands in an online manner and can lead to performance drop when incorporating with budget-forcing (Muennighoff et al., 2025). Recent Hybrid-CoT researches provide another perspective by reframing efficient reasoning as an implicit text classification problem. LRMs are trained to dynamically choose between thinking mode and non-thinking mode. Nevertheless, they typically rely on large-scale online RLVR (Zhang et al., 2025a; Tu et al., 2025). There are also some works explore offline tuning paradigms. For instance, DAST (Shen et al., 2025) designed a difficulty-based ranking function for preference and leveraged SimPO for training. Ada-R1 (Luo et al., 2025a) employed model merging followed by DPO on bi-level preference data to explicitly distinguish between two thinking modes.

## 3 METHOD

### 3.1 OVERVIEW

We refine the training procedure based on three key ingredients of RL: 1) Data: We harvest outputs from the generation space of the model using only one seamless rollout process and a small-scale prompt data. 2) Reward: Reward is implicitly determined by the ranking of data. We adopt an effective strategy to select data and rank purely by length to maximize the preference signal. 3) Algorithm: We investigate the objectives of existing preference optimization methods, and introduce Length Control Preference Optimization (LCPO) for more efficient length pruning.

### 3.2 DATA AND REWARD: EMPIRICAL STUDY ON GENERATION SPACE

In this section, we aim to answer RQ1: Is there still shorter yet effective reasoning paths within the generation space of LRMs? We conduct experiments to analyze the trajectory distribution of the generation space of LRMs. Throughout this process, we introduce our design on data and reward.

**Analysis Settings and Notations.** In this section, we conduct analysis on LIMR (Li et al., 2025) with DeepSeek-R1-Distill-Qwen-7B and DeepSeek-R1-Distill-Qwen-1.5B. Questions in LIMR are objectively difficult, but relatively easy for the model. For each question $q_i$ in the dataset $D$, we generate 16 outputs and sort them by length (e.g., the number of tokens in the sequence). This procedure samples a set of reasoning trajectories from the output space of model $M$:

$$\{(q_i, \{o_i^r\}_{r=1}^{16}, s_i)\}_{i=1}^{|D|}, \tag{1}$$

where $o_i^r$ is the r-th shortest output of $q_i$, $s_i$ denotes the average accuracy of question $q_i$ over $\{o_i^j\}_{j=1}^{16}$.

First, we evaluate how LRMs' performance across the whole dataset varies with the r-th shortest reasoning trajectory set $\{o_i^r\}_{i=1}^D$. As shown in Figure 2a, outputs with longer average lengths exhibit a noticeable performance decline: the 7B model shows degradation for rankings above 10, while the 1.5B model declines earlier from ranking 7. Conversely, for rankings 0-6 with shorter outputs, performance remains stable despite length variations. This suggests that reasoning models with extended deliberation tendencies can still perform efficient, shorter reasoning without significant accuracy loss, providing a positive answer to RQ1. This highlights the potential of leveraging model-generated data for behavior refinement.

Although model performance shows a clear decline with longer outputs, we argue that this is a correlation rather than a causal relationship. For reasoning-oriented models, their tendency to reflect and self-verify often leads to longer responses on questions they answer incorrectly, as they typically perform more extensive trial-and-error reasoning (Sui et al., 2025; Liu et al., 2025c). This suggests that output length is influenced by the model's perception of question difficulty, indicating that responses to more difficult questions may inherently contain more complex and lengthier generation mode. While the model can achieve over 90% accuracy with approximately 2200 tokens, it requires an average length of 7000 tokens to complete reasoning on the same tasks. This indicates a misalignment between the model's perceived difficulty and its actual capabilities. Even on tasks well within its capabilities, the model still wastes excessive tokens on reasoning due to its inflated sense of challenge. To this end, we adopt a coarse and heuristic approach to approximate question from the model's perspective and differentiate the conciseness of generated responses. Specifically, we categorize each question $q_i$ along with its outputs $o_i^r{}_{r=1}^{16}$ based on the corresponding score $s_i$:

$$\text{label}(q_i) = \begin{cases} \text{easy} & \text{if } s_i = 1, \\ \text{medium} & \text{if } 0 < s_i < 1, \\ \text{difficult} & \text{if } s_i = 0. \end{cases} \qquad (2)$$

To encourage the model to maintain exploratory behaviors when encountering difficult problems while correcting its perception to prevent excessive responses on simple tasks, we train on questions the model has already mastered. Thus, for data, we leverage the easy split of the generated data. And for reward (e.g. ranking of preference data), we select the shortest response as the chosen one and the longest as the rejected one. Details about the datasets refer to Table 7.

### 3.3 Algorithm: Empirical Study of Preference Optimization Methods

In this section, we aim to answer RQ2: How can we leverage limited training and data to adjust reasoning model's generation distribution? First, we provide brief preliminaries of preference optimization, which is further detailed in Appendix C. Then we begin with a preliminary experiment to investigate the potential of existing preference optimization methods in effectively capturing length preferences. After that, we leverage Bradley-Terry model (Bradley & Terry, 1952) to gain theoretical insight of the convergence characteristics of these methods. Finally, based on our findings and analysis, we introduce Length Controlled Preference Optimization (LCPO), an effective preference optimization method for shortening output length of reasoning models with small scale training.

**Bradley-Terry Model.** Let $\beta_i \in \mathbb{R}$ represent the ability value of a team $i$. Let the outcome of a game between teams $i$ and $j$ be determined by the difference of ability values $\beta_i - \beta_j$. The Bradley-Terry model (Bradley & Terry, 1952) defines the log-odds that team $i$ beats team $j$ as

$$\log \frac{p_{ij}}{1 - p_{ij}} = \beta_i - \beta_j, \; p_{ij} = \frac{1}{1 + e^{-(\alpha + \beta_i - \beta_j)}} = \sigma(\beta_i - \beta_j), \qquad (3)$$

**DPO and Preference Optimization** Let $x$ be the input sampled from the context space $\mathcal{X}$. We define an LLM $\theta$ as our policy $\pi_\theta(y|x)$, where the probability of generating a sequence $y$ is $\pi_\theta(y|x)$. Reinforcement Learning from Human Feedback (RLHF) (Ouyang et al., 2022) uses unpaired data. It adds a KL divergence (Kullback & Leibler, 1951) penalty to the reward to restrict the policy distribution from shifting excessively. The objective is to maximize accumulated rewards:

$$L_{\text{RLHF}} = -\mathbb{E}_{x \sim \mathcal{D}, \, y \sim \pi_\theta(x, y)} \big[ r(x, y) - \beta \, \mathbb{D}_{KL}(\pi_\theta(y|x) \,\|\, \pi_{\text{ref}}(y|x)) \big]. \qquad (4)$$

Table 2: Results of different preference optimization methods. We report accuracy (Acc) and average number of tokens in the generation (Len). **Bold** indicates the best trade-off. $\Delta \uparrow \%$ denotes length reduction ratio (higher is better).

| Method | MATH-500 | | GSM8K | |
|---|---|---|---|---|
| | Acc | Len ($\Delta \uparrow \%$) | Acc | Len ($\Delta \uparrow \%$) |
| Original (Guo et al., 2025) | 92.20 | 4223 | 91.81 | 1677 |
| SFT (Ouyang et al., 2022) | 91.00 | 3844 (9.97%) | 89.41 | 805 (52.00%) |
| DPO (Rafailov et al., 2023) | 92.00 | 2823 (33.15%) | 92.84 | 1059 (36.85%) |
| SimPO (Meng et al., 2024) | 91.80 | 3750 (11.20%) | 92.87 | 1687 (-5.96%) |
| ORPO (Hong et al., 2024) | 91.20 | 3137 (25.72%) | 91.43 | 988 (41.09%) |
| SimPER (Xiao et al., 2025) | 91.80 | 3832 (9.26%) | 92.19 | 1629 (2.86%) |
| Ours | **91.40** | **2033 (51.86%)** | **92.95** | **796 (52.53%)** |

Direct Preference Optimization (DPO) (Rafailov et al., 2023) derives an implicit reward from RLHF:

$$r_{\text{DPO}}(x, y) = \beta \log \frac{\pi_\theta(y|x)}{\pi_{\text{ref}}(y|x)} + \beta \log Z(x), \tag{5}$$

where $Z(x) = \sum_y \pi_{\text{ref}}(y|x) \exp(\frac{1}{\beta} r(x, y))$ is the partition function that is hard to estimate. DPO leverages the BT model to eliminate $Z(x)$, resulting in the following objective:

$$L_{\text{DPO}} = -\mathbb{E}_{x \sim \mathcal{D}, y \sim \pi_\theta(x,y)} [\log \sigma(\beta \log \frac{\pi_\theta(y_w|x)}{\pi_{\text{ref}}(y_w|x)} - \beta \log \frac{\pi_\theta(y_l|x)}{\pi_{\text{ref}}(y_l|x)})]. \tag{6}$$

### 3.3.1 DIFFERENCES AMONG PREFERENCE OPTIMIZATIONS

**Experimental Setup** We train DeepSeek-R1-Distill-Qwen-7B on the easy split of LIMR and evaluate different preference optimization methods on MATH-500 (Lightman et al., 2023) and GSM8K (Cobbe et al., 2021). Training configurations are provided in Appendix B.

Table 2 summarizes the results of our experiment. For the MATH-500 dataset, among existing methods, DPO (Rafailov et al., 2023) achieves superior performance with proper tuning on hyperparameters, reducing average output length by 33.15%. SFT (Ouyang et al., 2022) shows limited efficacy, yielding only a 9.97% length reduction. Notably, the recently proposed SimPER (Xiao et al., 2025) underperforms with merely 9.26% reduction. ORPO (Hong et al., 2024) demonstrates more significant improvement (25.72%), substantially outperforming SFT. On the GSM8K benchmark, SFT and ORPO exhibit an accuracy degradation but achieve more substantial length reductions (52.00% and 41.09% respectively).

### 3.3.2 INSIGHT OF CONVERGENCE CHARACTERISTICS ON PREFERENCES

Several approaches show clustered patterns. We further investigate where these convergence characteristics come from and how they influence preference alignment. We analyze the objective functions of multiple methods in a general setting, applying the insights to scenarios with length preferences.

We reformulate the objective functions of different methods into a log-sigmoid function form:

$$-\log \sigma(R(y_w, y_l, |x)) \tag{7}$$

The transformed objectives can then be interpreted as specialized BT models, where the sigmoid's parameters $R(y_w, y_l, |x)$ explicitly encode the reward difference between competing objective functions. During preference optimization, the model converges when the sigmoid output approaches 1, indicating near-deterministic preference satisfaction. Thus, we assess convergence by verifying the condition where $\sigma(R(y_w, y_l|x)) \to 1$. **Detailed derivations are provided in Appendix D**.

We theoretically illustrate in Equation 21 that the less satisfactory outcomes of SFT stem from the NLL loss. For ORPO, as shown in Inequation 26, as the probability of the chosen response rises, the convergence of the objective becomes increasingly dominated by the NLL loss. In Equation 31 we find that although SimPER can be helpful in non-reasoning tasks or relatively easy reasoning tasks

(e.g. GSM8K (Cobbe et al., 2021), as shown in (Xiao et al., 2025)), it is less effective in preference alignment for length control. DPO (Inequation 32) and SimPO (Inequation 33) have more relaxed convergence conditions but are highly dependent on hyperparameters, which can lead to unstable performance with limited training data.

### 3.3.3 AN OBJECTIVE BETTER FOR CONVERGENCE ON PREFERENCE DATA

Based on the analysis, we can conclude that two ingredients are needed in the context of length control: 1) Bradley-Terry loss with a relaxed reward margin and 2) well balanced negative rewards for the NLL loss. So first our objective follows the conventional BT loss form:

$$- \log \sigma(\beta r_\theta(y_w|x) - \beta r_\theta(y_l|x) + \epsilon). \tag{8}$$

We set $\epsilon$ to zero to let our model converge earlier for ingredient 1). Note that the reward of the NLL part can be shown explicitly in an objective of Bradley-Terry form:

$$L_{\text{NLL}} = -\frac{1}{|y|} \log \pi_\theta(y|x) = \log \sigma(\log \frac{p_\theta(y|x)}{1 - p_\theta(y|x)}) \tag{9}$$

where

$$p_\theta(y|x) = \exp(\frac{1}{|y|} \log \pi_\theta(y|x)), \ r_\theta(y|x) = log(\frac{p_\theta(y|x)}{1 - p_\theta(y|x)}) \tag{10}$$

We directly balance this reward with a negative counterpart for ingredient 2). Then our proposed objective is as follow:

$$L_{\text{LCPO}} = \log \sigma(\log(\frac{p_\theta(y_w|x)}{1 - p_\theta(y_w|x)}) - \log(\frac{p_\theta(y_l|x)}{1 - p_\theta(y_l|x)})). \tag{11}$$

Our objective can converge rapidly without requiring any hyperparameter tuning.

## 4 EXPERIMENTS

### 4.1 MAIN RESULT

**Experimental Setup**  We choose DeepSeek-R1-Distill-Qwen-7B and DeepSeek-R1-Distill-Qwen-1.5B (Guo et al., 2025) as base models.  We evaluate our method on the following math reasoning benchmarks: MATH-500 (Lightman et al., 2023), GSM8K (Cobbe et al., 2021), Minerva-Math (Lewkowycz et al., 2022), AIME24 (Online, 2025), AMC23 (of America, 2025) and OlympiadBench (He et al., 2024). We use accuracy as the metric for evaluation. Considering the limited size of AIME24 and AMC23, we sample 16 outputs for each question and compute pass@1 score (e.g., accuracy averaged over 16 runs). Information about the benchmarks and baselines can be found in Appendix A. Detailed experiment settings can be found in Appendix B.

**LCPO effectively reduces length while maintaining reasoning performance.**  As shown in Table 3, the experimental results demonstrate that our method achieves consistent effectiveness across diverse benchmarks. Our approach significantly reduces generation length by at least 46% across all datasets. On the challenging Minerva-Math, our method prunes 64% of the output, while bringing a slight performance gain. With our method, reasoning performance remains stable across most benchmarks, with only slight drops on a few.  Overall, our method achieves more than 50% average length reduction over all benchmarks and yields the best overall trade-off, while the computation demands are lower.

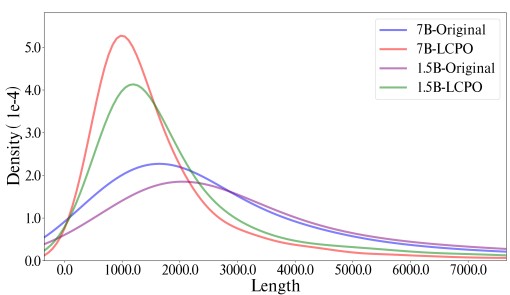

Figure 3: The distribution of output length on MATH-500 shifts after preference optimization.

Table 3: (Averaged) accuracy (Acc) and averaged number of tokens in generation (Len) of our method on different benchmarks. **Bold** denotes the best trade-off results. $\Delta$ denotes change on Acc. $\Delta\%$ denotes the change ratio of Len. *Total* denotes the total change on Acc. *Avg* denotes the average change ratio of Len.

| Method | MATH-500 Acc($\Delta$) Len($\Delta\%$) | GSM8K Acc($\Delta$) Len($\Delta\%$) | Minerva-Math Acc($\Delta$) Len($\Delta\%$) | AIME24 Acc($\Delta$) Len($\Delta\%$) | AMC23 Acc($\Delta$) Len($\Delta\%$) | OlympiadBench Acc($\Delta$) Len($\Delta\%$) | Total Avg |
|---|---|---|---|---|---|---|---|
| | | | DeepSeek-R1-Distill-Qwen-1.5B | | | | |
| Original | 83.00 5665 | 86.28 2457 | 26.84 7211 | 29.17 16355 | 70.62 10162 | 44.66 11715 | - - |
| CoD | 81.80 (-1.20) 4788 (-15.48%) | 80.89 (-5.39) 1488 (-39.44%) | 25.74 (-1.10) 6231 (-13.59%) | 26.67 (-2.50) 15465 (-5.44%) | 69.84 (-0.78) 9052 (-10.92%) | 43.03 (-1.63) 11014 (-5.98%) | -12.6 -15.14% |
| L1-Exact | 81.80 (-1.20) 3335 (-41.13%) | 87.95 (+1.67) 2986 (+21.53%) | 25.37 (-1.47) 3673 (-49.06%) | 21.25 (-7.92) 3665 (-77.59%) | 67.19 (-3.43) 3406 (-66.48) | 42.88 (-1.78) 3657 (-68.78%) | -14.13 -46.92% |
| L1-Max | 82.20 (-0.80) 3374 (-40.44%) | 88.32 (+2.04) 3141 (+27.84%) | 27.57 (+0.73) 3418 (-52.60%) | 22.29 (-6.88) 3523 (-78.46%) | **70.62 (0.00)** 3250 (-68.02%) | 43.03 (-1.63) 3452 (-70.53%) | -6.54 -47.04% |
| TrEff | 82.80 (-0.20) 2813 (-50.34%) | 83.24 (-3.64) 612 (-75.09%) | 25.74 (-1.10) 2950 (-59.10%) | 28.54 (-0.63) 10719 (-34.46%) | 69.84 (-0.78) 5166 (-49.16%) | 43.62 (-1.04) 7652 (-34.68%) | -7.39 -50.47% |
| Ours | **83.20 (+0.20)** 2397 (-57.69%) | **85.82 (-0.46)** 946 (-61.50%) | **27.21 (+0.37)** 2596 (-64.00%) | **29.58 (+0.41)** 8810 (-46.13%) | 70.47 (-0.15) 4026 (-60.38%) | **44.81 (+0.15)** 4921 (-57.99%) | **+0.52 -57.31%** |
| | | | DeepSeek-R1-Distill-Qwen-7B | | | | |
| Original | 92.20 4223 | 91.81 1677 | 36.76 5926 | 51.46 13411 | 87.97 6966 | 56.82 8789 | - - |
| CoD | 90.06 (-2.14) 2778 (-34.22%) | 88.32 (-3.49) 416 (-75.19%) | 36.40 (-0.36) 2733 (-53.88%) | 48.54 (-2.92) 12029 (-10.30%) | 87.34 (-0.63) 4880 (-29.95%) | 54.60 (-2.22) 7200 (-18.08%) | -11.76 -36.94% |
| L1-Exact | 89.80 (-2.40) 3555 (-15.82%) | 92.34 (+0.53) 3320 (+97.97%) | 36.03 (-0.73) 3341 (-43.62%) | 21.04 (-30.42) 3442 (-74.33%) | 69.53 (-18.44) 3337 (-52.10%) | 53.86 (-2.96) 3712 (-57.77%) | -54.42 -24.28% |
| L1-Max | 88.80 (-3.40) 2016 (-52.26%) | 92.42 (+0.61) 1640 (-2.21%) | 36.40 (-0.36) 1908 (-67.80%) | 25.00 (-26.46) 3560 (-73.45%) | 75.94 (-12.03) 3197 (-54.11%) | 54.30 (-2.52) 2510 (-65.14%) | -44.16 -52.50% |
| TrEff | 90.20 (-2.00) 2413 (-42.86%) | 89.39 (-2.42) 357 (-78.71%) | 37.13 (+0.37) 2917 (-50.78%) | 48.13 (-3.33) 10204 (-23.91%) | 87.03 (-0.94) 4420 (-36.55%) | 53.56 (-3.26) 6970 (-20.70%) | -11.58 -42.25% |
| DAST | 91.20 (-1.00) 3563 (-15.63%) | 91.21 (-0.60) 1092 (-34.88%) | 36.03 (-0.73) 8119 (+37.01%) | 50.83 (-0.63) 15430 (+15.05%) | 87.66 (-0.31) 6422 (-7.81%) | 55.34 (-1.48) 10339 (+17.64%) | -4.75 +11.38% |
| Ours | **91.40 (-0.80)** 2033 (-51.86%) | **92.95 (+1.14)** 796 (-52.53%) | **38.97 (+2.21)** 2079 (-64.92%) | 48.75 (-2.71) 6892 (-48.61%) | **86.88 (-1.09)** 3108 (-55.38%) | **56.08 (-0.74)** 4222 (-51.96%) | **-1.99 -54.21%** |

**Distribution is shifted by preference optimization.** Figure 3 shows the distribution of output length on MATH-500 before training and after training. It is clear that the peak of the distribution, which indicates the mean, is shifted left as expected. Meanwhile, after training, the variance is also decreased after training, resulting in a model with greater consistency on length preference and better alignment with short effective trajectories in the generation space.

## 4.2 ANALYSIS

We conduct additional experiments on the MATH-500 and GSM8K benchmarks to analyze key components of our pipeline. Based on the framework in 3.1, we investigate how the key ingredients (e.g., data with implicit reward and preference optimization algorithms) affect the effectiveness of our method on length reduction.

**Data with implicit reward.** Table 4 focuses on the data (with implicit reward), presenting the results of training on the split of different difficulty labels and the full dataset ("w/o filter"). As the length of the chosen responses in the training set increases, the average output length of models trained with our method also scales up, which aligns with our assumption that responses to harder problems contain more complex reasoning modes, and confirms the effectiveness of our data filtering strategy. Although all the chosen responses of the difficult split are not correct, the reasoning performance of the model is still slightly improved. This indicates that the LCPO objective focuses more on capturing preference and is robust to noise in the correctness label of training data.

Table 4: Results of training on splits of different difficulty labels. **Bold** indicates the best trade-off $\Delta \uparrow \%$ denotes length reduction ratio (higher is better).

| Method | MATH-500 | | GSM8K | | Avg. |
|---|---|---|---|---|---|
| | Acc | Len ($\Delta \uparrow \%$) | Acc | Len ($\Delta \uparrow \%$) | chosen Len |
| Original Model | 92.20 | 4223 | 91.81 | 1677 | - |
| Ours w/ easy | **91.40** | **2033 (51.86%)** | **92.95** | **796 (52.53%)** | 2232 |
| w/ medium | 91.80 | 2468 (41.56%) | 92.65 | 1068 (36.31%) | 3637 |
| w/ difficult | 92.00 | 3130 (25.88%) | 92.49 | 1364 (18.66%) | 3681 |
| w/o filter | 91.80 | 2954 (30.05%) | 92.34 | 1180 (29.64%) | 3270 |

Table 5: Results of our method on the out-of-distribution benchmarks. **Bold** denotes the best trade-off results. $\Delta$ denotes change on Acc. $\Delta\%$ denotes the change ratio of Len. *Total* denotes the total change on Acc. *Avg* denotes the average change ratio of Len.

| Method | MMLU | | GPQA-D | | Winogrande | | Total & Avg. |
|---|---|---|---|---|---|---|---|
| | Acc($\Delta$) | Len($\Delta\%$) | Acc($\Delta$) | Len($\Delta\%$) | Acc($\Delta$) | Len($\Delta\%$) | |
| | DeepSeek-R1-Distill-Qwen-1.5B | | | | | | |
| Original | 46.43 | 2549 | 34.85 | 10312 | 50.04 | 1516 | - |
| Ours | **46.78 (+0.35)** | **760 (-70.18%)** | **39.39 (+4.54)** | **4340 (-57.91%)** | **50.83 (+0.79)** | **309 (-79.62%)** | **+5.68 & -69.24%** |
| | DeepSeek-R1-Distill-Qwen-7B | | | | | | |
| Original | 65.27 | 1858 | 48.99 | 9237 | 61.17 | 1042 | - |
| Ours | **64.16 (-1.11)** | **835 (-55.06%)** | **52.53 (+3.54)** | **4301 (-53.44%)** | **62.12 (+0.95)** | **375 (-64.01%)** | **+3.38 & -57.50%** |

**Algorithm of Preference Optimization.** Table 2 also shows the performance of our objective comparing with various preference optimization methods. With limited training, LCPO achieves the best performance in reducing response length on MATH-500 while maintaining reasoning performance. Compared to SimPO, which is used in several previous works (Chen et al.; Shen et al., 2025), LCPO performs better at the early training stage. Moreover, LCPO's objective function requires no hyperparameter tuning, making it easier to use.

## 4.3 DISCUSSION

**Generalizability to OOD scenario.** To assess the effectiveness of our method to generalize in out-of-distribution scenarios, we conduct evaluations on MMLU (Hendrycks et al., 2021b;a), GPQA-Diamond (Rein et al., 2024) and WinoGrande (Sakaguchi et al., 2020) which are distinct from our training data in question form and subjects. These benchmarks likely fall outside our reasoning models' training distribution (Aggarwal & Welleck, 2025). As shown in Table 5, models trained on math datasets demonstrate remarkable generalization ability. On MMLU, the output length of the 1.5B and 7B models is reduced by over 70% and 55%, respectively, while the reasoning

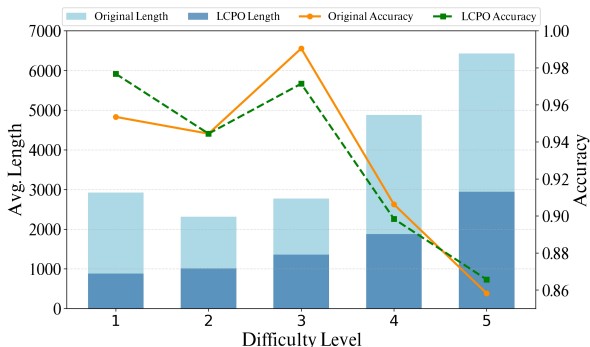

Figure 4: Changes of accuracy and generation length of DeepSeek-R1-Distill-Qwen-7B across different difficulty levels on MATH-500.

performance is maintained. On GPQA-Diamond, while the reduction for the two models exceeds 53%, the accuracy is improved by 3.54% and 4.54%, respectively. On average, we achieved a slight improvement in performance while significantly reducing the generation length.

**LCPO adaptively reduces generation length across different difficulty levels.** As shown in Figure 4, our method reduces over half of the generation length across different difficulty levels.

At the same time, accuracy of different levels is preserved. Intuitively, easier problem need less tokens to process, which should result in shorter generation on level 1 problems. However, Figure 4 shows a longer average length on level 1 problems than level 2 and level 3. This indicates a potential overthinking phenomenon of LRMs, where easier queries consume more tokens and can lead to performance drop (Liu et al., 2025b; Feng et al., 2025; Chen et al.; Sui et al., 2025). While our method reduces the generation length on level 1, accuracy on level 1 improves. After training, the average generation length is positively correlated with difficulty level, demonstrating consistency of reasoning effort with reasoning complexity.

## 5    CONCLUSION

In this paper, we investigated how to reduce the output length of large reasoning models via small-scale preference optimization. We proposed a pipeline for achieving trade-offs between reasoning performance and output length using limited data. We distilled data from models' generation space and filtered data with question-level difficulty estimation. We analyzed different preference optimization methods under a unified Bradly-Terry loss framework and proposed Length Controlled Preference Optimization (LCPO). Extensive experiments demonstrated that our method reduced output length substantially while maintaining reasoning performance. Our work highlights the potential of efficient methods for tuning model behaviors.

### ACKNOWLEDGMENTS

This research was partially supported by the National Science and Technology Major Project (No.2022ZD0117103), the National Natural Science Foundation of China (Grants No.62477044,62406303), Anhui Provincial Natural Science Foundation (No.2308085QF229), the Fundamental Research Funds for the Central Universities (No.WK2150110038), the Young Elite Scientists Sponsorship Program by CAST (No. 2024QNRC001), and the Meituan Joint Project.

### REPRODUCIBILITY STATEMENT

To facilitate reproduction and verification of our results, we provide the following details: (1) Implementation Details: Comprehensive implementation details, including hyperparameters, prompts, and algorithmic specifics, can be found in Appendix B. (2) Theoretical Analysis: The theoretical foundations and mathematical derivations supporting Section 3.3.2 are presented in Appendix C and Appendix D. (3) Source Code: https://github.com/SleepyWithoutCoffee/Small_Scale.

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

## A  BRIEF INTRODUCTION OF DATASETS AND BASELINES

**Math Datasets**  We use several math datasets covering in domain and out of domain data for evaluation. The detailed information is as follows:

- MATH-500 (Lightman et al., 2023) is a high-quality math problem solving dataset containing 500 problems extracted from the MATH (Hendrycks et al., 2021c) test set. It is widely used for evaluation of LLM reasoning (Guo et al., 2025; Yang et al., 2024b; Team, 2025).

- GSM8K (Cobbe et al., 2021) is a famous dataset containing 1,319 primary school level math problems. It is alse widely used for LLM evaluation (Ouyang et al., 2022; OpenAI, 2025; Liu et al.; 2025a).
- Minerva-Math (Lewkowycz et al., 2022) is a specialized collection designed for training and evaluating AI models on challenging mathematical reasoning tasks. It includes 272 problems ranging from algebra and calculus to advanced proofs. It is adopted by Luo et al. (2025b) for evaluation on LRMs trained via online RL.
- AIME24 (Online, 2025) is a collection of mathematical problems from the American Invitational Mathematics Examination (AIME) (Online, 2025) held in 2024. It contains 30 extremely challenging problems from the real world and is used for evaluating powerful LRMs.
- AIME25 (Online, 2025) is a another collection of 30 mathematical problems AIME (Online, 2025) held in 2025.
- AMC23 (of America, 2025) is a collection of challenging mathematical problems designed for the American Mathematics Competitions (AMC). It is also a widely used dataset.
- OlympiadBench (He et al., 2024) is an Olympiad-level bilingual multimodal scientific benchmark, featuring 8,476 problems from Olympic-level mathematics and physics competitions, including the Chinese college entrance exam. We use the math split in English for evaluation following previous works (Luo et al., 2025b; Aggarwal & Welleck, 2025).
- HLE (for AI Safety Phan Long agibenchmark@ safe. ai 1 Gatti Alice 1 Li Nathaniel 1 Khoja Adam 1 Kim Ryan 1 Ren Richard 1 Hausenloy Jason 1 Zhang Oliver 1 Mazeika Mantas 1 Hendrycks Dan dan@ safe. ai 1, 2026) is an extremely challenging multi-modal benchmark at the frontier of human knowledge,developed globally by subject-matter experts. Following Du et al. (2025), we use test data that does not require multimodal capabilities in the "Math" split, which is known as HLE-Math.

**General Datasets**  We adopt the widely used dataset MMLU (Hendrycks et al., 2021b), GPQA-Diamond (Rein et al., 2024) and WinoGrande (Sakaguchi et al., 2020) for evaluation.

- MMLU is a massive multitask test consisting of 14k multiple-choice questions from various branches of knowledge, spanning subjects in the humanities, social sciences, hard sciences, and other areas that are important for some people to learn. This covers 57 tasks including elementary mathematics, US history, computer science, law, and more. To attain high accuracy on this test, models must possess extensive world knowledge and problem solving ability.
- GPQA is a challenging dataset of 448 multiple-choice questions written by domain experts in biology, physics, and chemistry. Questions in GPQA are high-quality and extremely difficult, reaching human PhD level. GPQA-Diamond is a subset of GPQA containing 198 problems. It is the cleanest and most reliable part of the GPQA dataset and is widely used test set for reasoning.
- WinoGrande is a dataset of fill-in-a-blank commonsense reasoning problems, widely used to evaluate models' commonsense reasoning and general reasoning capabilities (Dubey et al., 2024; Team et al., 2025a;b).

**Coding Datasets**  We use the continuously updated benchmark, LiveCodeBench (Jain et al.).

- LiveCodeBench provides holistic and contamination-free evaluation of coding capabilities of LLMs. Particularly, LiveCodeBench continuously collects new problems over time from contests across competition platforms. We use the latest split v6 for evaluation.

**Training Datasets**  We use LIMR (Li et al., 2025) dataset for training. LIMR is a subset of the MATH (Hendrycks et al., 2021c) training set containing 1,389 math problems extracted by assessing consistency of accuracy reward scores during online RL. In our practice, we perform rollout on the whole dataset and use only 400 instances from it for training 50 steps.

**Baselines**  We evaluate baseline methods listed in Table 1. These baselines span across various types, covering inference-time, RL, RL-with-budget-forcing and preference optimization.

All the baselines are evaluated using the open-source models released by their respective authors. Detailed information about the baselines is provided below:

Table 6: Hyperparameters for various preference optimization methods. It is worth noting that the preference weight parameter is not required for LCPO.

| Method | learning rate | preference weight | others |
|---|---|---|---|
| SFT | 1e-5 | - | - |
| DPO | 5e-6 | 0.1 | - |
| SimPO | 5e-7 | 2 | $\gamma = 0.5$ |
| SIMPER | 5e-7 | - | - |
| ORPO | 5e-6 | 0.2 | - |
| LCPO (Ours) | 5e-6 | 0.3 | - |

- CoD (Xu et al., 2025b), Chain-of-Draft prompting, is a training-free inference-time method performed by adding an special instruction asking the model to keep a draft of at most 5 words for each thinking step.

- L1 (Aggarwal & Welleck, 2025) is an online RL based method incorporating with budget-forcing. The model is trained on a massive amount of data from the DeepScaleR dataset (Luo et al., 2025b). L1 provides two versions of model. L1-Exact is trained under an accurate length constrained reward penalty. L1-Max is trained under a ceiling limit of length constraint. We use the open-source weights from the official HuggingFace repository[1].

- TrEff (Arora & Zanette, 2025) is another powerful online RL based method. During training, TrEff adopts an additional normalized length reward similar to the advantage estimation of DeepSeek's GRPO (Shao et al., 2024). We use the open-source model released in the official repository[2].

- DAST (Shen et al., 2025) is a preference optimization based method. It uses pass rate to estimate difficulty of the questions and then calculate a token length budget. The a length penalty based on the budget is applied when ranking preference data to achieve adaptive slow thinking. The original paper uses DeepSeek-R1-Distill-Qwen-7B and DeepSeek-R1-Distill-Qwen-32B (Guo et al., 2025) as the base models. Their official repository[3] only contains the 7B model, which we adopt for evaluation.

## B  IMPLEMENTAION DETAILS

**Common Settings**  Experiments are conducted on $4\times$ A800-80G PCIe. We train the model for 3 epochs and save checkpoints for every 50 steps, using the checkpoint at the 50th step for all methods, which corresponds to only 0.4k sampled pairs. Methods in Table 2 also use the same common settings. For all experiments, we set batch size per GPU to 2, resulting in an equivalent batch size of 8. Context length is limited to 2.3k. The learning rate scheduler type is cosine, and the warmup ratio is 0.1. The optimizer is AdamW (Kingma, 2014) from torch (Paszke et al., 2017). The dtype is bfloat16.

For rollout and evaluation, we set temperature to 0.6. The max tokens limit is 32,768 in all experiments following Guo et al. (2025). For datasets with limited size (e.g., AIME24, AMC23), we report averaged accuracy over 16 samples (e.g. pass@1). Additionally, we report the average number of tokens as the average generation length which is denoted as *Len*.

**Hyperparameters**  The specific settings for the methods in Section 3.3 are listed in Table 6.

**Infrastructures**  Our training framework is built upon LLaMA-Factory (Zheng et al., 2024), while vLLM (Kwon et al., 2023) is employed for rollout.The evaluation pipeline is based on DeepScaleR (Luo et al., 2025b). Furthermore, we integrate official evaluation scripts from the repositories of various datasets.

**Prompts**  We use the recommended prompt setting from DeepSeek official repo (Guo et al., 2025), shown in Figure 5.

---

[1]https://huggingface.co/collections/l3lab/l1-67cacf4e39c176ca4e9890f4
[2]https://huggingface.co/daman1209arora/models
[3]https://github.com/AnonymousUser0520/AnonymousRepo01

---

**Evaluation Prompt for Math**

{Math Problem}

Please reason step by step, and put your final answer within \\boxed{}.

---

**Evaluation Prompt for General**

{General Problem}

Please reason step by step and put your final answer (eg, A, B, C, D) within \\boxed{}, such as "\\boxed{A}".

---

Figure 5: Evaluation prompt.

Table 7: Statistics of all the datasets generated from LIMR. Token Proportion (%) indicates the share of tokens contributed by each category to the overall total, which also reflects the allocation of GPU inference budget.

| Model | Split | Number of Questions | Token Proportion (%) |
|---|---|---|---|
| DeepSeek-R1-Distill-Qwen-7B | easy | 988 | 56.3 |
| DeepSeek-R1-Distill-Qwen-7B | medium | 330 | 38.3 |
| DeepSeek-R1-Distill-Qwen-7B | difficult | 52 | 5.4 |
| DeepSeek-R1-Distill-Qwen-7B | all | 1389 | 100 |
| DeepSeek-R1-Distill-Qwen-1.5B | easy | 488 | 19.4 |

**Generated Datasets** In this paper, we employ a heuristic approach involving self-distillation (rollout), filtering, and ranking of the data. Through this process, we obtain three splits: easy, medium, and difficult. Table 7 presents the statistics of the dataset generated via this pipeline. For the medium split, we select the shortest correct response as the chosen one and the longest wrong response as the rejected one. Furthermore, we filter out questions where the longest wrong response is shorter than the shortest correct answer. For the difficult split, we treat the shortest response as the chosen one and the longest as the rejected one.

## C INTRODUCION FOR PREFERENCE OPTIMIZATION

In this section, we outline the foundational concepts and notations used throughout our study of preference optimization, including the formulation of pairwise preference data, reward modeling, and the learning objectives employed in various algorithms such as DPO, SimPO, and others.

**Bradley-Terry Model.** Let $\beta_i \in \mathbb{R}$ represent the ability value of team $i$. Let the outcome of a game between teams $i$ and $j$ be determined by the difference in their aibility values $\beta_i - \beta_j$ . The Bradley-Terry model (Bradley & Terry, 1952) defines the log-odds corresponding to the probability $p_{ij}$ that team $i$ beats team $j$ as

$$\log \frac{p_{ij}}{1 - p_{ij}} = \alpha + \beta_i - \beta_j, \tag{12}$$

where $\alpha$ is an intercept term. Solving for $p_{ij}$ yields

$$p_{ij} = \frac{1}{1 + e^{-(\alpha + \beta_i - \beta_j)}} = \sigma(\alpha + \beta_i - \beta_j), \tag{13}$$

where $\sigma(\cdot)$ is the sigmoid function. In the LLMs setting, given an input $x$ within a finite space of contexts $\mathcal{X}$, we define a policy as $\pi_\theta(y|x)$. For a pair of actions $(y_i, y_j)$, we obtain a partial preference relation $y_i \succeq y_j$ between them through human annotations or similar methods, indicating $y_i$ is preferred over $y_j$ according to the ground truth. Tuning the policy for preference on a dataset

$\mathcal{D} = \{(x^{(i)}, y_w^{(i)}, y_l^{(i)})\}_{i=1}^{|\mathcal{D}|}$ aims to maximize the expected log-probability of $y_w^{(i)} \succeq y_l^{(i)}$. We take the rewards $r(x^{(i)}, y_w^{(i)})$ and $r(x^{(i)}, y_l^{(i)})$ of $y_w^{(i)}$ and $y_l^{(i)}$ given by a rule function or reward model as their ability values. Then applying the BT model gives

$$L(r, \mathcal{D}) = -\mathbb{E}_{(x,y_w,y_l)\sim\mathcal{D}}[\log \sigma(r(x, y_w) - r(x, y_l)]. \tag{14}$$

**RLHF and DPO.** Reinforcement Learning from Human Feedback (RLHF) (Ouyang et al., 2022) uses unpaired data generated by the policy $\pi_\theta(y|x)$. It adds a KL divergence penalty to the reward to restrict the policy distribution from shifting excessively. The objective is to maximize the accumulated rewards:

$$L(\theta) = -\mathbb{E}_{x\sim\mathcal{D},y\sim\pi_\theta(x,y)}[r(x, y) - \beta\mathbb{D}_{KL}(\pi_\theta(y|x)||\pi_{\text{ref}}(y|x)]. \tag{15}$$

Based on the RLHF formation, Direct Preference Optimization (DPO) (Rafailov et al., 2023) derives an implicit reward:

$$r_{\text{DPO}}(x, y) = \beta \log \frac{\pi_\theta(y|x)}{\pi_{\text{ref}}(y|x)} + \beta \log Z(x), \tag{16}$$

where $Z(x) = \sum_y \pi_{\text{ref}}(y|x) exp(\frac{1}{\beta} r(x, y))$ is the partition function that is hard to estimate. DPO leverage BT loss to remove $Z(x)$ from the objective, resulting in the folllowing objective:

$$L_{\text{DPO}} = -\mathbb{E}_{x\sim\mathcal{D},y\sim\pi_\theta(x,y)}[\log \sigma(\beta \log \frac{\pi_\theta(y_w|x)}{\pi_{\text{ref}}(y_w|x)} - \beta \log \frac{\pi_\theta(y_l|x)}{\pi_{\text{ref}}(y_l|x)})]. \tag{17}$$

**SimPO, SimPER and ORPO.** Simple Preference Optimization (SimPO) (Meng et al., 2024) introduces a reward margin and uses the average log probability of a sequence as the implicit reward to eliminate the need for the reference model in the DPO objective:

$$L_{\text{SimPO}} = -\mathbb{E}_{x\sim\mathcal{D},y\sim\pi_\theta(x,y)}[\log \sigma(\frac{\beta}{|y_w|} \log \pi_\theta(y_w|x) - \frac{\beta}{|y_l|} \log \pi_\theta(y_l|x) - \gamma)]. \tag{18}$$

Simple alignment with Perplexity optimization (SimPER) (Xiao et al., 2025) utilizes perplexity (Jelinek et al., 1977) and changes the form of BT loss into a simpler one:

$$L_{\text{SimPER}} = -\mathbb{E}_{x\sim\mathcal{D},y\sim\pi_\theta(x,y)}[\exp(\frac{1}{|y_w|} \log \pi_\theta(y_w|x)) + \exp(\frac{1}{|y_l|} \log \pi_\theta(y_l|x))]. \tag{19}$$

Odds Ratio Preference Optimization (ORPO) (Hong et al., 2024) incorporates an odds ratio penalty (formulated as a BT loss) into the conventional negative log-likelihood (NLL) loss:

$$L_{\text{ORPO}} = -\mathbb{E}_{x\sim\mathcal{D},y\sim\pi_\theta(x,y)}[\log p_\theta(y_w|x) - \lambda \log \sigma(\log \frac{p_\theta(y_w|x)}{1 - p_\theta(y_w|x)} - \log \frac{p_\theta(y_l|x)}{1 - p_\theta(y_l|x)})], \tag{20}$$

where $p_\theta(y|x) = \exp(\frac{1}{|y|} \log \pi_\theta(y|x))$.

We will leverage the definition of $p_\theta(y|x)$ for further derivation.

# D  ANALYSIS OF CONVERGENCE CHARACTERISTICS OF PREFERENCE OPTIMIZATION METHODS

We aim to find an ideal method of preference optimization that can achieve fast convergence to a good point in the context of length control. We begin by reformulating the objectives into the form of BT loss. And then we analyze the convergence conditions of different optimization objectives based on practical assumptions.

**Background Assumptions From Practice** For sequences with length $|y| > 1000$, the typical per-token log-probabilities lie in the range $\log p(y_t) \in [-5, 0]$. Let $m$ be the point where the sigmoid function saturates. Let $\sigma(m) > 0.99$, then we have $m > 5$. We make an assumption about the training dynamics: after updating, a convergence of the model on preference satisfies $p_\theta(y_w \mid x) > p_\theta(y_l \mid x)$ by some margin, where $y_w$ and $y_l$ denote the preferred and less-preferred responses, respectively. Note that this could lead to over-fitting if it holds for every sample. However, this assumption is reasonable when considering the average case over the input distribution $\mathcal{X}$ (e.g. *it is satisfied for a set of inputs with high probability mass*).

Table 8: Objective functions of preference optimization methods (expectation omitted).

| Name | Objective |
|---|---|
| SFT (Ouyang et al., 2022) | $-\frac{1}{|y_w|} \log \pi_\theta(y_w|x)$ |
| DPO (Rafailov et al., 2023) | $-\log \sigma \left( \beta \log \frac{\pi_\theta(y_w|x)}{\pi_{\text{ref}}(y_w|x)} - \beta \log \frac{\pi_\theta(y_l|x)}{\pi_{\text{ref}}(y_l|x)} \right)$ |
| SimPO (Meng et al., 2024) | $-\log \sigma \left( \frac{\beta}{|y_w|} \log \pi_\theta(y_w|x) - \frac{\beta}{|y_l|} \log \pi_\theta(y_l|x) - \gamma \right)$ |
| ORPO (Hong et al., 2024) | $-\log p_\theta(y_w|x) - \lambda \log \sigma \left( \log \frac{p_\theta(y_w|x)}{1-p_\theta(y_w|x)} - \log \frac{p_\theta(y_l|x)}{1-p_\theta(y_l|x)} \right),$ where $p_\theta(y|x) = \exp \left( \frac{1}{|y|} \log \pi_\theta(y|x) \right)$ |
| SimPER (Xiao et al., 2025) | $-\exp \left( \frac{1}{|y_w|} \log \pi_\theta(y_w|x) \right) + \exp \left( \frac{1}{|y_l|} \log \pi_\theta(y_l|x) \right)$ |

**SFT**  With some algebra, we can derive the BT form of SFT loss as:

$$L_{\text{SFT}} = -\log \sigma(\log \frac{p_\theta(y_w|x)}{1 - p_\theta(y_w|x)}), \tag{21}$$

where the penalty for the rejected responses is absent, or $p_\theta(y_l|x)$ is default to 0.5. Convergence achieved when:

$$\frac{p_\theta(y_w|x)}{1 - p_\theta(y_w|x)} > e^m \Rightarrow p_\theta(y_w|x) > \frac{e^m}{1 + e^m}. \tag{22}$$

For $m = 5(\sigma \approx 0.993)$, requires $p_\theta(y_w|x) > 0.993$. For sequences longer than 1000, cumulative probability requires:

$$\prod_{t=1}^{1000} p(y_w^{(t)}|x, y_w^1, ..., y_w^{t-1}) > 0.993. \tag{23}$$

This is fundamentally incompatible with typical token probabilities ($p \approx 0.05 - 0.5$). Thus SFT cannot effectively enforce preference alignment, which aligns with intuition. In another words, the convergence of SFT depends completely on the input $x$. In the context of length control, fitting the model to limited generation modes leads to potential catastrophic forgetting.

**ORPO**  For ORPO, it is complex due to the additional NLL loss.

$$L_{\text{ORPO}} = -\log \sigma[\log(\frac{1 + (1 + e^{-z})p_\theta(y_w|x)^{1/\lambda}}{e^{-z} - (1 + e^{-z})p_\theta(y_w|x)^{1/\lambda}})], \tag{24}$$

where $z = \log \frac{p_\theta(y_w|x)}{1-p_\theta(y_w|x)} - \log \frac{p_\theta(y_l|x)}{1-p_\theta(y_l|x)}$. Then we have:

$$2p_\theta(y_w|x)^{1/\lambda} > \frac{p_\theta(y_l|x) - p_\theta(y_w|x)}{p_\theta(y_l|x) + p_\theta(y_w|x)} + m \tag{25}$$

Continue the derivation:

$$\log(1 + (1 + \exp(-z))p_\theta(y_w|x)^{\frac{1}{\lambda}}) - \log(\exp(-z) - (1 + \exp(-z))p_\theta(y_w|x)^{\frac{1}{\lambda}}) > m \tag{26}$$

$$1 + (1 + \exp(-z))p_\theta(y_w|x)^{\frac{1}{\lambda}} > \exp(m - z) - \exp(m)(1 + \exp(-z))p_\theta(y_w|x)^{\frac{1}{\lambda}} \tag{27}$$

$$2p_\theta(y_w|x)^{\frac{1}{\lambda}} > \frac{\exp(m - z) - 1}{(1 + \exp(m - z))(1 + \exp(m))} = \frac{1}{1 + \exp(m)}(1 - \frac{2}{1 + \exp(m - z)}) \tag{28}$$

$$2p_\theta(y_w|x)^{\frac{1}{\lambda}} > \frac{1}{1 + \exp(m)} \cdot (1 - 2\sigma(z - m)) \tag{29}$$

Note that $z$ is inside the sigmoid function of the BT loss from the original objective. Then $z - m$ controls whether the right side is positive. If $z - m > 0$ (e.g, $p_\theta(y_w|x) > p_\theta(y_l|x) + \text{some\_margin}$), the right side is negative, which satisfies the inequality. Then the convergence behavior is determined

by the SFT loss component. In this case, we can apply the analysis of SFT. If $z - m < 0$ (e.g, $p_\theta(y_w|x) < p_\theta(y_l|x) + \text{some\_margin}$), then we have

$$\frac{\text{sf}}{1 + \exp(m)}^{\frac{1}{\lambda}} < p_\theta(y_w|x) < p_\theta(y_l|x) + \text{some\_margin}, \tag{30}$$

where sf is a positive scaling factor smaller than 1. Based on the assumption that $p_\theta(y_w|x) > p_\theta(y_l|x)$ *by some margin is satisfied for a set of inputs with high probability mass*. We consider the latter case as under-fit. In conclusion, ORPO fits better and faster to the preference data than SFT. **However, due to the NLL component in its objective, it fits worse than pure log odds ratio loss.**

**SimPER**   SimPER cannot be written as a log sigmoid formation. Because the value of log sigmoid function always falls in $(-\infty, 0)$, while the inverse of the SimPER objective is always positive under the assumption that $p_\theta(y_l|x) < p_\theta(y_w|x)$ *by some margin is satisfied for a set of inputs with high probability mass*. This implies that the convergence of SimPER is unconstrained by preference. From the formulation of SimPER's objective function, it can be observed that it separately optimizes two objectives of the following form:

$$p_\theta(y|x) = \exp\left(\frac{1}{|y|} \log \pi_\theta(y|x)\right) > 0, \tag{31}$$

which cannot be written as a log sigmoid formation either. Thus, the convergence of SimPER depends on both responses.

**DPO and SimPO**   These two methods have similar objective. We can directly extract the difference of the rewards from their objectives. For DPO we have:

$$\log \pi_\theta(y_w|x) - \log \pi_\theta(y_l|x) > \log \frac{\pi_{\text{ref}}(y_w|x)}{\pi_{\text{ref}}(y_l|x)} + \frac{m}{\beta} \tag{32}$$

For SimPO we have:

$$\frac{1}{|y_w|} \log \pi_\theta(y_w|x) - \frac{1}{|y_l|} \log \pi_\theta(y_l|x) > \frac{\gamma + m}{\beta} \tag{33}$$

Intuitively, the normalized advantage objective in SimPO potentially provides superior convergence properties for long-form generation tasks, while DPO remains valuable for maintaining response diversity. The convergence characteristics depend on the hyperparameters settings, which decide the right sides of the equation. Note that the left sides can be seen as the reward differences of DPO and SIMPO. Thus, the right sides indicate reward margin floor.

**Conclusion**   From the performance and analysis of SFT, ORPO and SimPER, we can conclude that Bradley-Terry loss provide predictable convergence characteristics from the objective, while NLL loss potentially amplifies the data's impact on convergence. Under the form of BT loss, the tunable component of the objective is the reward function. Thus, a negative reward is needed to mitigate the effect of NLL. Besides, from the anlysis of DPO and SimPO, we can conclude that the reward margin is important in length control, which is reflected directly in the effect controlled by the hyperparameters.

# E  EXTENDED RELATED WORKS: HYBRID CHAIN OF THOUGHT

Chain-of-Thought (Wei et al., 2022b) is intended to guide models to employ effective reasoning patterns in solving complex problems through prompt (Xue et al., 2024; Zhao et al.). To achieve performance breakthroughs, foundation models undergo post-training that enables them to autonomously apply complex reasoning patterns inspired by CoT during generation Yang et al. (2024a); Guo et al. (2025); Yang et al. (2024b); Ma et al. (2025).

Hybrid Chain-of-Thought or Adaptive Thinking aims to enable LLMs to only apply complex reasoning patterns on challenging prompts. LLMs with adaptive thinking ability dynamically choose between two reasoning modes based on problem difficulty: (1) highly complex thinking with thinking tags and non-empty thinking content, or (2) non-thinking without thinking tags or with empty

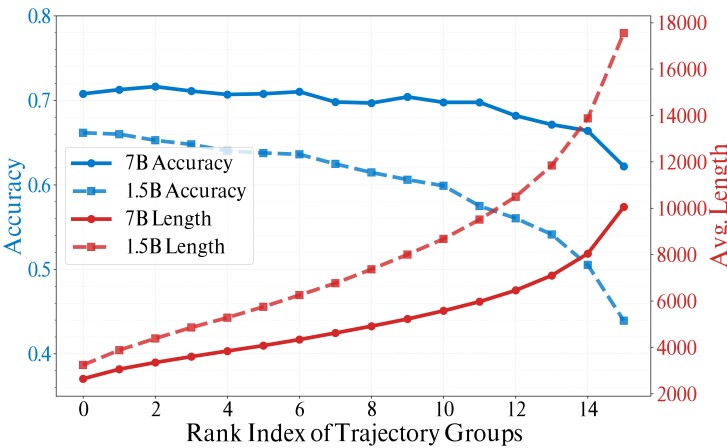

Figure 6: A replicated experiment of Section 3 on Eurus-2 dataset (Cui et al., 2025; Yuan et al., 2024).

Table 9: (Averaged) accuracy (Acc) and averaged number of tokens in generation (Len) of our method on OOD benchmarks. **Bold** denotes the best trade-off results. Δ denotes change on Acc. Δ% denotes the change ratio of Len. *Total* denotes the total change on Acc. *Avg* denotes the average change ratio of Len.

| Method | MMLU | | GPQA-D | | WinoGrande | | |
|---|---|---|---|---|---|---|---|
| | Acc(Δ) | Len(Δ%) | Acc(Δ) | Len(Δ%) | Acc(Δ) | Len(Δ%) | Total & Avg. |
| | | | DeepSeek-R1-Distill-Qwen-1.5B | | | | |
| Original | 46.43 | 2549 | 34.85 | 10312 | 50.04 | 1516 | - |
| CoD | 46.72 (+0.29) | 1592 (-37.54%) | 33.33 (-1.52) | 7328 (-28.94%) | 46.88 (-3.16) | 1572 (+3.69%) | -4.39 & -20.93% |
| L1-Exact | 48.64 (+2.21) | 5373 (+52.56%) | 31.31 (-3.54) | 3253 (-68.45%) | 51.14 (+1.10) | 4819 (+217.88%) | -0.23 & +67.33% |
| L1-Max | 49.29 (+2.86) | 2219 (-12.95%) | 31.82 (-3.03) | 2839 (-72.47%) | 50.59 (+0.55) | 2409 (+58.91%) | +0.38 & -8.84% |
| TrEff | 44.80 (-1.63) | 1201 (-52.88%) | 27.27 (-7.58) | 6914 (-32.95%) | 49.09 (-0.92) | 795 (-47.56%) | -10.13 & -44.46% |
| Ours | **46.78 (+0.35)** | **760 (-70.18%)** | **39.39 (+4.54)** | **4340 (-57.91%)** | **50.83 (+0.79)** | **309 (-79.62%)** | **+5.68 & -69.24%** |
| | | | DeepSeek-R1-Distill-Qwen-7B | | | | |
| Original | 65.27 | 1858 | 48.99 | 9237 | 61.17 | 1042 | - |
| CoD | 63.72 (-1.55) | 822 (-55.76%) | 47.98 (-1.01) | 5834 (-36.84%) | 62.12 (+0.95) | 505 (-51.54%) | -1.61 & -48.05% |
| L1-Exact | 63.77 (-1.50) | 2834 (+52.53%) | 46.46 (-2.53) | 3235 (-64.98%) | 62.90 (+1.73) | 2949 (+64.67%) | -2.30 & +17.41% |
| L1-Max | 64.02 (-1.25) | 983 (-47.09%) | 46.97 (-2.02) | 2039 (-77.93%) | 62.19 (+1.02) | 847 (-18.71%) | -2.25 & -47.91% |
| TrEff | 63.17 (-2.10) | 1077 (-42.03%) | 44.95 (-4.04) | 6470 (-29.96%) | 60.22 (-0.95) | 472 (-54.70%) | -7.09 & -42.43% |
| DAST | 65.62 (+0.35) | 2468 (+32.83%) | 47.98 (-1.01) | 10819 (+17.13%) | 59.35 (-1.82) | 1630 (+56.43%) | -2.48 & +35.46% |
| Ours | **64.16 (-1.11)** | **835 (-55.06%)** | **52.53 (+3.54)** | **4301 (-53.44%)** | **62.12 (+0.95)** | **375 (-64.01%)** | **+3.38 & -57.50%** |

thinking content. Many concurrent works explore this paradigm for reducing the average generation length of LRMs. For instance, Ada-R1 (Luo et al., 2025a) used model merging and bi-level preference optimization to explicitly teach the model to distinguish between the two types of data. AdaptThink (Zhang et al., 2025a) carefully designed a reward function so that when facing data of different difficulties, the model's behaviors adaptively converge to a state mixing both modes as training progresses. AutoThink (Tu et al., 2025) adopted multi-stage RL to enable the model to master this output pattern. Although there are offline methods like Ada-R1, many other works such as AdaptThink and AutoThink still rely on costly large-scale online RL.

# F COMPLETE OOD RESULTS

Our method maintains strong generalization in out-of-distribution scenarios and achieves the best trade-off between reasoning capability and generation length, as shown in Table 9.

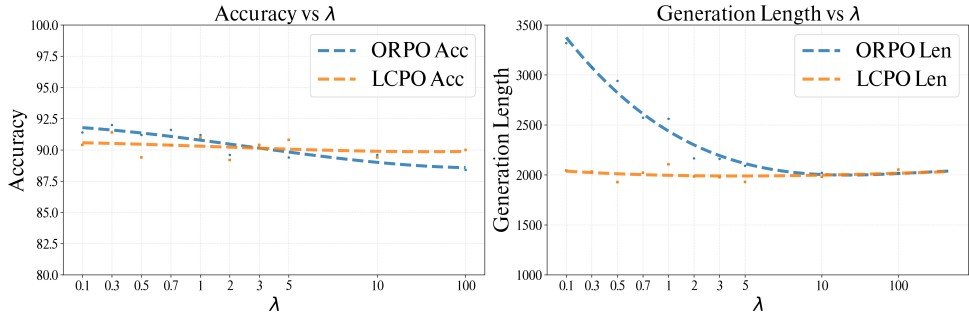

Figure 7: Comparison between ORPO and LCPO on MATH-500.

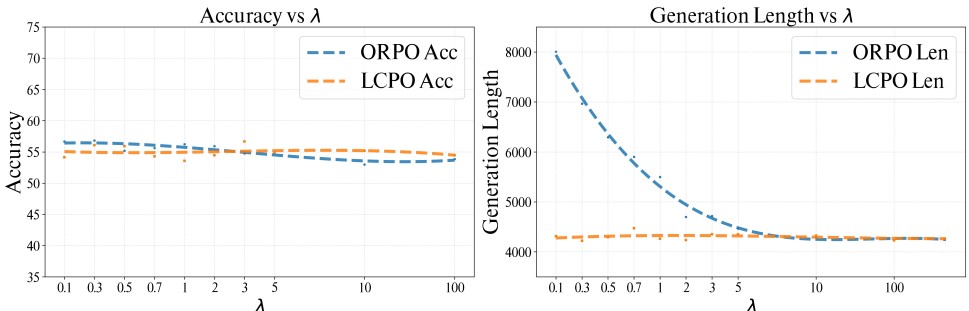

Figure 8: Comparison between ORPO and LCPO on OlympiadBench.

## G  ADDITIONAL RESULTS

### G.1  REPLICATED EXPERIMENTS OF SECTION 3.2

To verify whether the phenomena observed in Section 3.2 persists on more challenging datasets, we reran the experiment with identical settings on a more challenging dataset, the AMC-AIME split from Eurus-2 (Cui et al., 2025; Yuan et al., 2024).

The results in Figure 6 show that when confronted with harder problems, the model's performance and length still exhibit similar patterns and short effective paths still exist.

### G.2  SENSITIVITY ANALYSIS OF LCPO

To investigate how the weight of the BT loss (denoted as $\lambda$ following ORPO) affects LCPO, we compared the performance variations between LCPO and ORPO under different BT loss weighting configurations. We tested weights ranging from [0.1, 0.3, 0.5, 0.7, 1, 2, 3, 5, 10, 100], using the 7B model on MATH-500 and OlympiadBench. Results are shown in Figure 7 and 8.

**Sensitivity Analysis**  LCPO is robust to $\lambda$, making it a **hyperparameter-free** preference optimization method. It can match the peak performance of ORPO without any hyperparameter tuning or calculation on additional loss term.

As shown in the results, ORPO is sensitive to $\lambda$. When the optimization of ORPO becomes dominated by SFT, which is not always beneficial for reasoning (Chu et al.), a notable manifestation is that as the impact of the BT loss diminishes, the model overfits to the chosen responses. This could lead to a collapse in diversity and potentially causes a subsequent decline in accuracy (as shown in the figure, ORPO's accuracy gradually falls below that of LCPO as $\lambda$ increases).

In contrast, LCPO mitigates the effect by balancing the implicit NLL term. As the BT weight increases, the performance of ORPO continues to improve. When $\lambda$ becomes extremely large ($> \approx$ 10), it gradually approaches LCPO, which is predictable: the two loss terms in ORPO act together

Table 10: Performance of our method on LiveCodeBench (v6), AIME25 and HLE-Math.

| Method | LiveCodeBench (v6) | | AIME25 | | HLE-Math | |
|---|---|---|---|---|---|---|
| | Acc ($\Delta$) | Len ($\Delta\%$) | Acc ($\Delta$) | Len ($\Delta\%$) | Acc ($\Delta$) | Len ($\Delta\%$) |
| Original-1.5B | 18.29 | 16340 | 22.71 | 15985 | 3.07 | 14940 |
| Ours-1.5B | **18.86 (+0.57)** | **7871 (-51.83%)** | **22.23 (-0.48)** | **7194 (-56.00%)** | **3.48 (+0.41)** | **6217 (-58.39%)** |
| Original-7B | 33.14 | 12080 | 37.50 | 14622 | 4.30 | 13956 |
| Ours-7B | **33.71 (+0.57)** | **8018 (-33.63%)** | **36.87 (-0.63)** | **8187 (-44.01%)** | **4.51 (+0.21)** | **5873 (-57.92%)** |

Table 11: Cost of 7B model estimated in A100*h

| Method | Type | GPU hours |
|---|---|---|
| AdaptThink | RL | $\sim$1792 |
| AutoThink | RL | $\sim$5760 |
| TrEff | RL | $\sim$560 |
| Ours | preference optimization | $\sim$10.4 |

through summation. When the weight of one term increases, its influence and gradually surpasses that of the other and becomes dominant. This aligns with our analysis in Appendix D: as preference learning progresses, the probability of the chosen response gradually becomes higher than that of the rejected response by a certain margin. At this point, the BT loss no longer affects convergence, and the effect of the loss function is almost entirely dominated by SFT. Moreover, this indicates that LCPO directly achieves the theoretically optimal position by removing SFT, which reinforces our key insight: the dominant driver of performance in this paradigm is the BT loss part. This also aligns with the principle of Occam's Razor.

### G.3 ADDITIONAL EVALUATION

Coding is another important part of reasoning (Jiang et al.). We conducted additional evaluation on LiveCodeBench v6 split (the latest split) to assess the coding performance of our method. As shown in Table 10, our method also performs well on challenging code generation tasks. Acc improves slightly by 0.57, while Len is reduced by 33.63% and 51.83% for the 7B and the 1.5B models, respectively.

We also conducted additional evaluation on the new challenging math benchmarks AIME25 and HLE-Math. Following Du et al. (2025), we use an LLM (Guo et al., 2025) with their evaluation prompt to verify the generated answer for the HLE-Math evaluation. As shown in Table 10, our method also performs well. Notably, on HLE-Math Acc improves slightly by 0.41 and 0.21, while Len is reduced by 58.39% and 57.92% for the 1.5B and the 7B models, respectively.

### G.4 PIPELINE COST

The rollout process is the primary computational cost of our method. Here, we estimate the overall training cost (rollout and training) of the 7B model in A100*h[4] in Table 11.

**The rollout is a one-time, offline process.** Once the preference dataset is generated, it can be reused for multiple training runs and easily shared (e.g., via platforms like HuggingFace).

**The overall cost of our method is low.** Compare to offline methods (in Table 1), we need less data from rollout. Compare to online RL methods, which necessitate a continuous and interleaved rollout and policy updating throughout the entire training process, the overall computational budget (rollout + training) of our method remains highly competitive.

---

[4]Due to conversion discrepancies in GPU hours, the specific computational cost may fluctuate, but they can provide a reference in terms of magnitude.

Table 12: Results on generation diversity of our method.

| Model | Distinct-1 ↑ | Distinct-2 ↑ | Distinct-3 ↑ | EAD ↑ |
|---|---|---|---|---|
| Original-7B | 0.0222 | 0.1132 | 0.2320 | 0.0256 |
| Ours-7B | **0.0292** | **0.1260** | **0.2375** | **0.0312** |
| Original-1.5B | 0.0199 | 0.1036 | 0.2179 | 0.0240 |
| Ours-1.5B | **0.0285** | **0.1316** | **0.2553** | **0.0313** |
| Original-7B (on level 5) | 0.0046 | 0.0252 | 0.0546 | 0.0058 |
| Ours-7B (on level 5) | **0.0062** | **0.0303** | **0.0613** | **0.0070** |
| Original-1.5B (on level 5) | 0.0040 | 0.0222 | 0.0498 | 0.0055 |
| Ours-1.5B (on level 5) | **0.0063** | **0.0322** | **0.0668** | **0.0074** |

Table 13: Generation diversity comparisons.

| Model | Distinct-1 ↑ | Distinct-2 ↑ | Distinct-3 ↑ | EAD ↑ |
|---|---|---|---|---|
| Original-7B | 0.0222 | 0.1132 | 0.2320 | 0.0256 |
| ORPO-7B | 0.0275 | 0.1220 | 0.2326 | 0.0297 |
| Ours-7B | **0.0278** | **0.1253** | **0.2400** | **0.0300** |
| Original-7B on level 5 | 0.0046 | 0.0252 | 0.0546 | 0.0058 |
| ORPO-7B on level 5 | 0.0059 | 0.0294 | 0.0597 | 0.0067 |
| Ours-7B on level 5 | **0.0060** | **0.0301** | **0.0614** | **0.0068** |

### G.5 GENERATION DIVERSITY

We conducted experiments to verify the impact of our method on generation diversity. To quantitatively measure the diversity of generations, we adopted the following metrics:

- Distinct-n (Li et al., 2016; Wang et al.): a metric used to calculate the diversity of a group of sentences,

- Expectation-Adjusted-Distinct (EAD) (Liu et al., 2022): an improved Distinct metric that removes the biases of the original Distinct score (Distinct tends to assign lower scores to longer sequences),

and test our method on MATH-500, which was chosen for its diverse and annotated difficulty levels. We generated 16 responses for each problem using the same generation settings as Table 3. We report results on both the full dataset and the most difficult problems (Level 5, the highest difficulty level) to quantitatively measure the diversity of generations on average and on difficult problems. As shown in Table 12, the model trained with our method achieves a higher score under all four metrics. This indicates that our method has improved generation diversity while simultaneously reducing overall response length.

We set $\lambda$ to 10 (where ORPO approaches LCPO in length reduction) and conduct a similar experiment. As shown in Table 13, the model trained with our method still achieves higher scores under all four metrics.

### G.6 HYBRID CoT

We further compared our method with AdaptThink and AutoThink, which have publicly available open-source weights. Results are shown in Table 11 (cost) and Table 14 (performance and generation length). Both methods serve as very strong baselines. Yet, our approach consistently demonstrates comparable performance with a much lower cost.

## H LIMITATIONS AND FUTURE WORK

We discuss several limitations of our work in this section. 1) Following previous works (Aggarwal & Welleck, 2025; Shen et al., 2025), our training set focuses on math tasks. The diversity of concise

Table 14: (Averaged) accuracy (Acc) and averaged number of tokens in generation (Len) of our method on different benchmarks. **Bold** denotes the best trade-off results. $\Delta$ denotes change on Acc. $\Delta\%$ denotes the change ratio of Len. *Total* denotes the total change on Acc. *Avg* denotes the average change ratio of Len.

| Method | MATH-500 Acc($\Delta$) Len($\Delta\%$) | GSM8K Acc($\Delta$) Len($\Delta\%$) | Minerva-Math Acc($\Delta$) Len($\Delta\%$) | AIME24 Acc($\Delta$) Len($\Delta\%$) | AMC23 Acc($\Delta$) Len($\Delta\%$) | OlympiadBench Acc($\Delta$) Len($\Delta\%$) | Total Avg |
|---|---|---|---|---|---|---|---|
| DeepSeek-R1-Distill-Qwen-1.5B | | | | | | | |
| Original | 83.00 5665 | 86.28 2457 | 26.84 7211 | 29.17 16355 | 70.62 10162 | 44.66 11715 | - - |
| AdaptThink | 80.20 (-2.80) 1715 (-69.73%) | 82.64 (-3.64) 447 (-81.81%) | 25.00 (-1.84) 1910 (-73.51%) | 27.08 (-2.09) 8494 (-48.07%) | 68.28 (-2.34) 3188 (-68.63%) | 41.84 (-2.82) 4309 (-63.22%) | -15.83 -67.50% |
| AutoThink | 81.00 (-2.00) 1820 (-67.87%) | 78.98 (-6.30) 601 (-75.54%) | 27.57 (+0.73) 2397 (-66.76%) | 28.96 (-0.21) 7338 (-55.13%) | 67.34 (-3.28) 3487 (-65.69%) | 44.36 (-0.30) 4049 (-65.44%) | -11.36 -66.07% |
| LCPO | 83.20 (+0.20) 2397 (-57.69%) | 85.82 (-0.46) 946 (-61.50%) | 27.21 (+0.37) 2596 (-64.00%) | 29.58 (+0.41) 8810 (-46.13%) | 70.47 (-0.15) 4026 (-60.38%) | 44.81 (+0.15) 4921 (-57.99%) | **+0.52** **-57.31%** |
| DeepSeek-R1-Distill-Qwen-7B | | | | | | | |
| Original | 92.20 4223 | 91.81 1677 | 36.76 5926 | 51.46 13411 | 87.97 6966 | 56.82 8789 | - - |
| AdaptThink | 90.60 (-1.60) 1975 (-53.23%) | 91.28 (-0.53) 354 (-78.89%) | 36.40 (-0.36) 2787 (-52.97%) | 53.54 (+2.08) 11002 (-17.96%) | 86.56 (-1.41) 4701 (-32.52%) | 55.93 (-0.89) 6594 (-24.97%) | -2.71 -43.42% |
| AutoThink | 89.40 (-2.80) 2187 (-48.21%) | 86.81 (-5.00) 809 (-51.76%) | 36.03 (-0.73) 2794 (-52.85%) | 50.42 (-1.04) 8215 (-38.74%) | 83.91 (-4.06) 4005 (-42.51%) | 56.97 (-0.15) 4817 (-45.19%) | -13.78 -46.54% |
| LCPO | 91.40 (-0.80) 2033 (-51.86%) | 92.95 (+1.14) 796 (-52.53%) | 38.97 (+2.21) 2079 (-64.92%) | 48.75 (-2.71) 6892 (-48.61%) | 86.88 (-1.09) 3108 (-55.38%) | 56.08 (-0.74) 4222 (-51.96%) | **-1.99** **-54.21%** |

reasoning behaviors depends on the native potential of LRMs. While experiments demonstrate the generalization ability to OOD scenarios, we believe there can be better results by carefully design a mixed training set of different reasoning modes. 2) Due to limited computational resources, we only conduct experiments on 1.5B and 7B models. Nevertheless, these extensive experiments demonstrate the effectiveness of our method across different model size.

## I  THE USE OF LARGE LANGUAGE MODELS (LLMS)

We use LLMs purely for writing assistance. Specifically, we employ LLMs to check and correct potential grammatical errors, adjust wording, and streamline the text to avoid exceeding the page limit. The LLMs we use are all freely accessible to the public: DeepSeek-V3.1[5] and ChatGPT[6].

---

[5]https://chat.deepseek.com/
[6]https://chatgpt.com/

