# OpenReview forum: "Pruning Long Chain-of-Thought of Large Reasoning Models via Small-Scale Preference Optimization"
_ICLR.cc/2026/Conference — ICLR 2026 Poster_

### Official Review · Reviewer_jTBW · 2025-10-31

**Soundness:** 2
**Presentation:** 3
**Contribution:** 2
**Rating:** 4
**Confidence:** 3

**Summary:**

This paper presents LCPO, a lightweight method for pruning overly long CoT outputs from large reasoning models. The authors mine short yet effective paths via self-distillation and difficulty filtering, then bias the policy toward them with a preference objective. On six math benchmarks, LCPO cuts average output length by >50% while preserving or slightly improving accuracy. Extensive results on OOD tests (MMLU, GPQA-D) show that the approach is general.

**Strengths:**

1. The paper reframes length reduction as “mine the model’s own short paths, then align”, and LCPO’s single-parameter BT loss is reasonable.
2. Extensive baselines (SFT, DPO variants) and OOD tests (MMLU, GPQA-D) are included.
3. With 0.8 k pairs and 50 steps, LCPO method can reduce average output length by over 50%.

**Weaknesses:**

1. Algorithmic novelty is limited.

    LCPO amounts to dropping the NLL term of ORPO and keeping the BT loss; a controlled ablation that simply scales up the BT weight in ORPO is missing, so the reader cannot tell whether the observed gains require a new objective or just a different hyper-parameter balance.

2. Data efficiency.

    The entire dataset must be rolled out 16× to obtain per-question accuracy and retain only the “easy” split. Moreover, all training pairs are self-generated, and the paper does not explore whether distilled high-quality pairs from another model (i.e., OOD preference data) could improve compression or robustness.

3. Hyper-parameter sensitivity

    The manuscript reports only the best value of the key temperature λ, without any sensitivity analysis.

**Questions:**

1. Algorithmic novelty:

   Have you run an ablation where the BT-loss weight in ORPO is increased? If so, please report the length/accuracy curve and indicate whether LCPO still outperforms this baseline.

2. Data efficiency:

    What fraction of the total GPU budget is spent on rolling out medi um/difficult questions that are finally discarded? Could a two-stage or cheaper proxy be used to predict “easy” questions and reduce upfront compute?

3. OOG Geeralization:

    Have you tried training LCPO on short–long pairs distilled from another model (e.g., GPT-4o) instead of pure self-distillation?

---

> ### Author Response · Authors · 2025-11-23
> **Rebuttal by Authors (Part 1/4)**
>
> Dear Reviewer jTBW,
>
> Thank you very much for your thorough review and constructive feedback! In our response, we have carefully addressed your comments with the following discussions.
>
> We have categorized the issues mentioned in your comments into three parts and respond to them one by one.
> > *Algorithmic novelty & Hyper-parameter sensitivity*
> >
> > *LCPO amounts to dropping the NLL term of ORPO and keeping the BT loss; a controlled ablation that simply scales up the BT weight in ORPO is missing, so the reader cannot tell whether the observed gains require a new objective or just a different hyper-parameter balance. The manuscript reports only the best value of the key temperature λ, without any sensitivity analysis. Have you run an ablation where the BT-loss weight in ORPO is increased? If so, please report the length/accuracy curve and indicate whether LCPO still outperforms this baseline.*
>
> Thank you for the insightful comment! We followed your valuable suggestions and conducted additional experiments. Since the BT loss weight can be shared between the two different objectives, **we present the results of both the ablation study and the hyperparameter sensitivity analysis in the same chart**. But first, we would like to clarify our unique contribution and the fundamental differences between our method and ORPO. Then, we demonstrate and analyze the results, based on which we further illustrate the advantages of our method.
>
> Our objective function resembles ORPO in mathematical from. However, our core contribution is **not merely the empirical finding that  LCPO is beneficial for length preference alignment in efficient reasoning**. Instead, it is the **theoretical insight and derivation** that led us to this specific, simplified, and more effective objective (Section 3.3.3) and the whole systematic pipeline for efficient reasoning powered by LCPO. We began with an empirical investigation into the effectiveness of different preference optimization methods for effective reasoning on small-scale training data. Then we gained insight into the actual inhibitory effect of NLL loss in this scenario under an unified BT loss framework.
>
> While **we aim to balance the effect of NLL loss, the core idea of ORPO is to introduce a minor penalty term based on log odds ratio into the SFT objective (or NLL loss), thereby achieving a form of preference-aligned SFT.** This is an intuitive design in direct contrast with our motivation. Furthermore, ORPO was originally proposed and validated in the context of general instruction-following tasks, without being extended to (efficient) reasoning domains where SFT (with NLL loss) is not necessary and can be detrimental to rapid preference alignment on length. However, **our discussion and design of preference optimization are directly targeted at (efficient) reasoning scenarios.** Therefore the resulting LCPO objective is **not an arbitrary term dropping but a principled design grounded in our theoretical understanding**.
>
> Results on MATH-500: **Figure 7 in Appendix F, from the revised PDF**
>
> Results on OlympiadBench: **Figure 8 in Appendix F, from the revised PDF**
>
> From the results we can draw the following conclusions:
>
> 1. Sensitivity analysis. **LCPO is robust to $\lambda$, making it a hyperparameter-free preference optimization method.** It can match the peak performance of ORPO without any hyperparameter tuning or calculation on additional loss term.
>
> 2. Comparison to ORPO.
>
>    2.1 **LCPO perform consistently better than ORPO.** As the BT weight increases, the performance of ORPO continues to improve. When λ becomes extremely large (>~10), it gradually approaches LCPO, which is predictable: the two loss terms in ORPO act together through summation. When the weight of one term increases, its influence naturally gradually surpasses that of the other and becomes dominant. This aligns with our analysis in Appendix C: as preference learning progresses, the  probability of the chosen response gradually becomes higher than that of the rejected response by a certain margin. At this point, the BT loss no longer affects convergence, and the effect of the loss function is almost entirely dominated by SFT. Therefore it does not imply that the only difference between the two lies in hyperparameter tuning. Moreover, this indicates  that **LCPO directly achieves the theoretically optimal position by removing SFT.** It would, in fact, reinforce our key insight: **the dominant driver of performance in this paradigm is the BT loss part**. This also aligns with the principle of Occam's Razor.

---

> ### Author Response · Authors · 2025-11-23
> **Rebuttal by Authors (Part 2/4)**
>
> 2.2 **ORPO is sensitive to $\lambda$, and excessive SFT compromises the model's generative diversity.** When the optimization of ORPO becomes dominated by SFT, which is not always beneficial for reasoning [5], a notable manifestation is that as the impact of the BT loss diminishes, the model overfits to the chosen responses. This could lead to a collapse in diversity and potentially causes a subsequent decline in accuracy (as shown in the figure, ORPO's accuracy gradually falls below that of LCPO as $\lambda$ increases). In contrast, LCPO mitigates this effect by balancing the implicit NLL term.
>
> To quantitatively measure the diversity of generations, we adopted the following metrics:
>
> Distinct-n [6, 7]: a metric used to calculate the diversity of a group of sentences,
>
> Expectation-Adjusted-Distinct (EAD) [8]: an improved Distinct metric that removes the biases of the original Distinct score (Distinct tends to assign lower scores to longer sequences),
>
> and conducted additional experiments on MATH-500, which was chosen for its diverse and annotated difficulty levels. We generated 16 responses for each problem with a **$\lambda$ of 10** (where ORPO approaches LCPO in length reduction). Our results are as follows:
>
> |                       | Distinct-1 $\uparrow$ | Distinct-2 $\uparrow$ | Distinct-3 $\uparrow$ | EAD $\uparrow$ |
> | :-------------------: | :-------------------: | :-------------------: | :-------------------: | :------------: |
> |      Original-7B      |        0.0222         |        0.1132         |        0.2320         |     0.0256     |
> |        ORPO-7B        |        0.0275         |        0.1220         |        0.2326         |     0.0297     |
> |        Ours-7B        |      **0.0278**       |      **0.1253**       |      **0.2400**       |   **0.0300**   |
> | Original-7B on level5 |        0.0046         |        0.0252         |        0.0546         |     0.0058     |
> |   ORPO-7B on level5   |        0.0059         |        0.0294         |        0.0597         |     0.0067     |
> |   Ours-7B on level5   |      **0.0060**       |      **0.0301**       |      **0.0614**       |   **0.0068**   |
>
> As shown in the table above, **the model trained with our method achieves a higher score under all four metrics**. This indicates that our method has improved generation diversity while simultaneously reducing overall response length.
>
> > *Data efficiency.*
> >
> > *The entire dataset must be rolled out 16× to obtain per-question accuracy and retain only the “easy” split. What fraction of the total GPU budget is spent on rolling out medium/difficult questions that are finally discarded? Could a two-stage or cheaper proxy be used to predict “easy” questions and reduce upfront compute?*
>
> Thank you for your review! Here are our responses to the questions.
>
> **The entire dataset must be rolled out 16× to obtain per-question accuracy and retain only the “easy” split.**
>
> We elaborate point by point on the rationale behind this design.
>
> Rolling out for 16x is a common setting of related works (for instance, in Table 1, L1 choose 16x). Therefore, our choice of the same number of rollouts is a fair and justified comparison. As mentioned in Section 3, the purpose of rollout is to sample from the model’s output trajectory distribution, thereby approximately reconstructing the trajectory space. **So the choice of 16 rollouts is not a strict requirement and can be flexibly increased or decreased depending on the budget.** Theoretically, a larger number of rollouts leads to a more comprehensive reconstruction of the trajectory space, and the resulting short effective trajectory length approaches the lower bound more closely. Considering both efficiency and effectiveness, we chose 16 rollouts. This number is also a common choice in other sampling-related works beyond efficient reasoning [0].
>
> Next, we justify our decision to retain only the "easy" split. For AI-generated synthetic data, it is often difficult to effectively control data  quality during the construction process. Therefore, performing data filtering after generation is a common and important practice for downstream tasks in LLM post-training [2]. In our scenario, as mentioned in Section 3 (line 215), the output length can be influenced by the model's perception of problem difficulty: it tends to generate more tokens when it perceives the  reasoning as challenging. We aim to prevent the model from overly perceiving difficulty in problems it has already mastered (labeled as "easy", but objectively may not be easy). Thus, **the model is encouraged to avoid excessive generation on the "easy" split only**. This serves as an effective data filtering strategy.
>
> Finally, the overall pipeline, including rollouts, remains efficient. The rollout is a one-time, offline process. Here are some intuitive numerical comparisons:

---

> ### Author Response · Authors · 2025-11-23
> **Rebuttal by Authors (Part 3/4)**
>
> 1. As detailed in Table 1 (column "Data") of our paper, our method requires less data from rollout generation (22k trajectories) compared to strong RL baselines like L1 (645k) and TrEff (24.8k) (where all the generated data are discarded).
> 2. For the 7B model training, TrEff spent $\sim$560 A100\*h for training the 7B model, while our method spent $\sim$10.4 A100\*h in total. Furthermore, since the generated data and the difficulty estimation results is reusable, no additional time is consumed for data rollout during repeated experiments.
>
> **Fraction of the total GPU budget is spent on rolling out medium/difficult questions.**
>
> For a 7B model, 43.7% of the rollout time is spent on generating tokens for medium/difficult questions.
>
> **Could a two-stage or cheaper proxy be used to predict “easy” questions and reduce upfront compute?**
>
> Thank you for raising this interesting thought! In this work, we aim to prevent the model from overly perceiving difficulty in problems it has already mastered. Therefore, decoupling the labeling of "easy" instances during data generation requires predicting the model’s performance in advance without relying on the model’s own generative process. However, a two stage difficulty estimation methods may not be reliable in our scenario.
>
> To address your question, we reviewed some existing literature on model performance prediction. Unfortunately, to the best of our knowledge, predicting model performance still requires sampling the model’s generated outputs in advance to train a predictor. Moreover, such predictors may not generalize well to new models or new datasets, which contradicts your expectation.
>
> As mentioned earlier, the overall pipeline of our method, including rollouts, remains efficient. Predicting "easy" questions is not central to our method. So this concern does not undermine our main contribution. However, we do believe your idea of a cheaper proxy for estimating model-specific difficulty is highly insightful and valuable. We leave it for future works.
>
> > *OOD Generalization.*
> >
> > *All training pairs are self-generated, and the paper does not explore whether distilled high-quality pairs from another model (i.e., OOD preference data) could improve compression or robustness. Have you tried training LCPO on short–long pairs distilled from another model (e.g., GPT-4o) instead of pure self-distillation?*
>
> Thank you for your insightful comment!
>
> While distilling knowledge from other models is indeed an interesting topic, we would like to clarify that it is not the core question we intend to investigate in our paper. **Our work explores how to achieve efficient reasoning through small-scale training.** We did not adopt cross model distillation for the following reasons:
>
> 1. **Efficient reasoning is an emerging research area. Unlike many general scenarios, there is no accessible teacher model (such as GPT-4o) that can directly provide high-quality distillation data.** Large reasoning models are generally trained through RLVR and exhibit a phenomenon known as test-time scaling, a positive correlation between output length and reasoning performance. Many powerful large-scale reasoning models, such as Deepseek-R1 [0], produce remarkably long outputs. On one hand, distillation from them may not be more efficient than self-distillation in terms of cost, which contradicts your expectation; on the other hand, it is also challenging to distill shorter reasoning trajectories from such models.  For non-reasoning powerful models (such as the GPT-4o you mentioned), their performance on challenging mathematical tasks lags behind that of the distillation models used in our experiments, making it difficult to obtain effective trajectories. Below are the results of GPT-4o on MATH-500 (reference from [1]).
>
> |             model             | MATH-500 |
> | :---------------------------: | :------: |
> |            GPT-4o             |   76.9   |
> |  GPT-4o + Majority Voting@5   |   78.9   |
> | DeepSeek-R1-Distill-Qwen-1.5B |   83.0   |
> |  DeepSeek-R1-Distill-Qwen-7B  |   92.2   |
>
> 2. **Aligning the model with its intrinsic patterns is helpful in the context of efficient reasoning.** In the filtering of training data, we considered the difficulty of questions from the model's perspective. Different models may perceive the difficulty of the same question differently and may generate patterns of short effective reasoning trajectories in distinct ways. This could potentially lead the model to align with a suboptimal preference for itself during subsequent training. Therefore, it is essential to align the distribution of training data with the model's own distribution.

---

> ### Author Response · Authors · 2025-11-23
> **Rebuttal by Authors (Part 4/4) and References**
>
> The primary goal of our work is to achieve rapid preference alignment on output length using limited data. The method we propose is a systematic pipeline, of which self-distillation is one highly effective component. **It is a practical setting, rather than a drawback.** However, your comment has brought great insight. We leave the development of a general distillation-based method for efficient reasoning as future work.
>
>
>
>
>
> References:
>
> [0] Guo, Daya, et al. "Deepseek-r1 incentivizes reasoning in llms through reinforcement learning." *Nature* 645.8081 (2025): 633-638.
>
> [1] Liu, Chengwu, et al. "Safe: Enhancing Mathematical Reasoning in Large Language Models via Retrospective Step-aware Formal Verification." *arXiv preprint arXiv:2506.04592* (2025). (Accepted by ACL'25)
>
> [2] Ye, Yixin, et al. "Limo: Less is more for reasoning." *arXiv preprint arXiv:2502.03387* (2025). (Accepted by COLM'25)
>
> [3] Zhang, Qiyuan, et al. "Collaborative Performance Prediction for Large Language Models." *Proceedings of the 2024 Conference on Empirical Methods in Natural Language Processing*. 2024.
>
> [4] Anugraha, David, et al. "Proxylm: Predicting language model performance on multilingual tasks via proxy models." *Findings of the Association for Computational Linguistics: NAACL 2025*. 2025.
>
> [5] Chu, Tianzhe, et al. "SFT Memorizes, RL Generalizes: A Comparative Study of Foundation Model Post-training." *Forty-second International Conference on Machine Learning*.
>
> [6] Li, Jiwei, et al. "A diversity-promoting objective function for neural conversation models." *Proceedings of the 2016 conference of the North American chapter of the association for computational linguistics: human language technologies*. 2016.
>
> [7] Wang, Chaoqi, et al. "Beyond Reverse KL: Generalizing Direct Preference Optimization with Diverse Divergence Constraints." *The Twelfth International Conference on Learning Representations*.
>
> [8] Liu, Siyang, et al. "Rethinking and Refining the Distinct Metric." *Proceedings of the 60th Annual Meeting of the Association for Computational Linguistics (Volume 2: Short Papers)*. 2022.

---

> ### Author Response · Authors · 2025-11-28
>
> Dear Reviewer,
>
> We sincerely appreciate your time and effort in reviewing our work and providing valuable feedback. Your insights have been instrumental in helping us improve the manuscript.
>
> In response to your comments, we have made our best efforts to address each point comprehensively. For open-ended questions, we have also provided thorough responses based on our technical understanding and survey conducted promptly during the rebuttal period.
>
> We have revised our manuscript based on the rebuttal. The specific modifications we made in the revised PDF are as follows:
>
> **1. Clarified the insensitivity to hyperparameter $\lambda$ in Section 3.3.3;**
>
> **2. Introduced hyperparameter sensitivity analysis experiments for LCPO and comparative experiments between LCPO and ORPO in Appendix F.2;**
>
> As the author-reviewer discussion period is drawing to a close, we kindly wish to confirm whether there remain any unresolved concerns or points requiring further clarification. We would be delighted to engage in deeper discussions to ensure all aspects are thoroughly addressed.
>
> Thank you again for your thoughtful review. We greatly value your expertise and look forward to your feedback.
>
> Best regards,
>
> The Authors

---

### Official Review · Reviewer_og6S · 2025-11-01

**Soundness:** 3
**Presentation:** 3
**Contribution:** 2
**Rating:** 6
**Confidence:** 4

**Summary:**

This paper proposes LCPO, a Length Controlled Preference Optimization, to implement efficient reasoning. Through extensive experimental analysis, the paper demonstrates the length difference in the model's own output distribution and the feasibility of using this difference for preference alignment. Furthermore, the paper experimentally analyzes the limitations of the method based on NLL loss. Finally, the authors propose using the log NLL loss as a reward for preference alignment, which achieves good results in length preference alignment.

**Strengths:**

1.The paper empirically demonstrates that short and correct reasoning trajectories exist within the model's generation distribution, which can be leveraged to model length preference.
2.The proposed LCPO method is highly effective and successfully alleviates the "overthinking" problem in large reasoning models.
3.The paper provides a good insight by analyzing the limitations of the NLL loss from an empirical perspective， and, based on this, proposes using a reward function derived from the NLL loss to model preferences.

**Weaknesses:**

1. Generalizability of In-Distribution Preference Alignment: While effective, the method of filtering and training on model-generated trajectories is highly dependent on the base model's intrinsic ability to produce short reasoning paths. This significantly limits the method's generalizability across different models and its potential for further compressing reasoning length.

2. Stability of LCPO: Although the authors claim the method is effective in small-scale data scenarios ()()()(), the lack of a reference model in LCPO raises concerns about distribution drift. It is difficult to guarantee that the model will not suffer from significant distribution shift, potentially leading to performance collapse, when trained on a larger volume of data. In this light, the reliance on small-scale data might be a necessary limitation rather than an advantage. Did the authors attempt to train the model on more data?

3. Limited Baselines: The paper primarily discusses methods for in-distribution length preference alignment (like DPO, SimPO, etc.) . It lacks a comprehensive comparison and discussion against other families of methods, such as broader RL-based approaches or prompt-based techniques.

**Questions:**

See above.

---

> ### Author Response · Authors · 2025-11-23
> **Rebuttal by Authors (Part 1/2)**
>
> Dear Reviewer og6S,
>
> Thank you very much for your thorough review and constructive feedback! In our response, we have carefully addressed your comments with the following discussions.
> > *1. Generalizability of In-Distribution Preference Alignment: While effective, the method of filtering and training on model-generated trajectories is highly dependent on the base model's intrinsic ability to produce short reasoning paths. This significantly limits the method's generalizability across different models and its potential for further compressing reasoning length.*
>
> Thank you for your insightful comment! We are delighted to engage in this discussion with you on this topic.
>
> **Aligning the model with its intrinsic patterns is necessary and helpful in the context of efficient reasoning, rather than a drawback.** In the filtering of training data, we considered the difficulty of questions from the model's perspective. Different models may perceive the difficulty of the same question differently and may generate patterns of short effective reasoning trajectories in distinct ways. This could potentially lead the model to align with a suboptimal preference for itself during subsequent training. Therefore, it is essential to align the distribution of training data with the model's own distribution, which is also a common setting of related works.
>
> The currently popular RL methods in the field of LLM reasoning have achieved remarkable results by tapping into the model's intrinsic ability [3, 4]. Although this approach cannot expand the model’s reasoning capabilities [3], a significant portion of the potential of pre-trained models may still remain untapped. Additionally, due to the costs associated with obtaining annotated trajectories (such as expenses for human annotation or API fees for proprietary models), methods based on the model's intrinsic ability may be more applicable in some scenarios.
>
> Efficient reasoning is an emerging research area. Unlike many general scenarios, there is no accessible teacher model (such as GPT-4o) that can directly provide high-quality distillation data. Due to differences in training methods and data, the range of generation lengths may vary across reasoning models, which makes aligning all reasoning models toward a unified length-preference standard extremely challenging. We leave the development of a cross-model general training method for efficient reasoning as future work.
>
> Once again, thank you for your forward-looking and highly insightful comment!
>
> > *2. Stability of LCPO: Although the authors claim the method is effective in small-scale data scenarios ()()()(), the lack of a reference model in LCPO raises concerns about distribution drift. It is difficult to guarantee that the model will not suffer from significant distribution shift, potentially leading to performance collapse, when trained on a larger volume of data. In this light, the reliance on small-scale data might be a necessary limitation rather than an advantage. Did the authors attempt to train the model on more data?*
>
> Thank you for your insightful review!
>
> Our work explores how to achieve efficient reasoning through small-scale training. For other methods (such as the baselines listed in Table 1 of this paper), due to their inherent characteristics, larger-scale data is often necessary to achieve stronger performance (for instance, RL-based methods typically require a substantial amount of data). In such cases, large-scale data is essential. However, our approach already demonstrates effectiveness with small-scale data, making further training on larger datasets unnecessary. This is inconsistent with our motivation.
>
> Your point regarding the absence of a reference model is very valuable. The reference model directly constrains the model from effectively learning the preference dataset. Therefore, there is a trade-off between preference alignment and staying close to the initial model [0]. In the context of RL, such trade-offs are also sometimes made to enhance the effectiveness of model training [1]. The primary goal of our method is to achieve rapid preference alignment on length with limited data. Hence, we choose not to use a reference model, similar to SimPO [0].
>
> In some scenarios, it may be necessary to incorporate more data to enhance specific capabilities of the model. In such cases, one can make the following practical adjustments:
>
> 1. Learning rate: a smaller learning rate naturally constrains model updates from the initial distribution.
> 2. In our objective, the implicit reward related to NLL loss ($r_{w} = \log(\frac{p_{\theta}(y_{w}|x)}{1-p_{\theta}(y_{w}|x)})$) is explicitly balanced to accelerate preference learning. One can introduce additional weight parameter(s) to increase the ratio of $r_{w}/r_{l}$, thereby mitigating the "squeezing effect"* [2] of negative gradients.
>
> This is a truly insightful comment! We will try to develop a more robust method towards efficient reasoning in future works.

---

> ### Author Response · Authors · 2025-11-23
> **Rebuttal by Authors (Part 2/2) and References**
>
> ****
>
> *"Squeezing effect" [2]: the decreased probability mass is largely “squeezed” into the output which was most confident before the update. A possible consequence is generating repeated phrases.
>
>
>
> > *3. Limited Baselines: The paper primarily discusses methods for in-distribution length preference alignment (like DPO, SimPO, etc.) . It lacks a comprehensive comparison and discussion against other families of methods, such as broader RL-based approaches or prompt-based techniques.*
>
> Thank you for your comment! Our work regards offline preference optimization as an efficient training scheme, focusing on how to reduce the model’s output length based on limited data and achieve a favorable trade-off between reasoning capability and output length. This is why we center our discussion on algorithm around preference optimization.
>
> As for other approaches: the main distinctions among different RL-based methods lie in the design of reward functions or reward models related to output length, while prompt-based methods tend to be more empirical. These factors make theoretical analysis of such methods challenging.
>
> In this paper, **we have already include these types of methods (as the following table shows).** We categorize the baseline methods **in Table 1** and compare them in subsequent experiments (Table 3, as well as Table 5, which will be extended in the revision).
> |     Baselines     |          Type           |
> | :---------------: | :---------------------: |
> |        CoD        |         prompt          |
> | L1-Exact & L1-Max |           RL            |
> |       TrEff       |           RL            |
> |       DAST        | preference optimization |
>
>
> References:
>
> [0] Meng, Yu, Mengzhou Xia, and Danqi Chen. "Simpo: Simple preference optimization with a reference-free reward." *Advances in Neural Information Processing Systems* 37 (2024): 124198-124235.
>
> [1] Yu, Qiying, et al. "Dapo: An open-source llm reinforcement learning system at scale." *arXiv preprint arXiv:2503.14476* (2025). (Accepted by NIPS'25)
>
> [2] Ren, Yi, and Danica J. Sutherland. "Learning Dynamics of LLM Finetuning." *The Thirteenth International Conference on Learning Representations*.
>
> [3] Yue, Yang, et al. "Does reinforcement learning really incentivize reasoning capacity in llms beyond the base model?." *arXiv preprint arXiv:2504.13837* (2025). (Accepted by NIPS'25)
>
> [4] Guo, Daya, et al. "Deepseek-r1 incentivizes reasoning in llms through reinforcement learning." *Nature* 645.8081 (2025): 633-638.

---

> ### Author Response · Authors · 2025-11-28
>
> Dear Reviewer,
>
> We sincerely appreciate your time and effort in reviewing our work and providing valuable feedback. Your insights have been instrumental in helping us improve the manuscript.
>
> In response to your comments, we have made our best efforts to address each point comprehensively. We have also revised our manuscript based on the rebuttal. Please let us know if there are any other modifications or supplementary materials you would like us to include in the revised manuscript.
>
> As the author-reviewer discussion period is drawing to a close, we kindly wish to confirm whether there remain any unresolved concerns or points requiring further clarification. We would be delighted to engage in deeper discussions to ensure all aspects are thoroughly addressed.
>
> Thank you again for your thoughtful review. We greatly value your expertise and look forward to your feedback.
>
> Best regards,
>
> The Authors

---

> > ### Comment · Reviewer_og6S · 2025-11-28
> >
> > Thanks for the response. After reading the replies, I believe the original rating was reasonable.

---

> > > ### Author Response · Authors · 2025-11-28
> > >
> > > We sincerely thank you for your time in reading our replies and responding to our rebuttal.
> > >
> > > Your comments significantly helped us to improve our work.
> > >
> > > We apologize for any inconvenience and truly appreciate your insightful feedback.
> > >
> > > Thank you again for your reply! Wishing you a pleasant weekend.

---

> > > ### Author Response · Authors · 2025-12-01
> > >
> > > Dear Reviewer og6S,
> > >
> > > We have updated our response to your comment W3. While drafting the summary, we realized that our previous reply might have led you to believe that we did not compare prompt-based and RL-based methods. To clarify, we have directly listed the types of baseline methods we employed to demonstrate that these categories are indeed covered in our work. As noted in the response, Table 1 in our paper outlines the types of methods covered, and extensive experimental comparisons of these methods are provided in Table 3 of the original PDF and Table 10 of the revised PDF.
> > >
> > > We regret that due to some issues with OpenReview, you are now unable to continue providing valuable feedback. Nonetheless, out of respect, we still wish to inform you of our response.

---

### Official Review · Reviewer_4RRh · 2025-11-01

**Soundness:** 3
**Presentation:** 3
**Contribution:** 2
**Rating:** 4
**Confidence:** 4

**Summary:**

The paper first analyzes the generation path distributions of large reasoning models and filters their generated trajectories based on difficulty estimation to identify concise yet effective reasoning patterns.
Then, it proposes Length Controlled Preference Optimization (LCPO) — a preference optimization method that directly balances implicit NLL-related rewards, effectively shortening reasoning chains by over 50% while maintaining accuracy.

**Strengths:**

1. This paper conducts a rigorous theoretical analysis of the characteristics of the Bradley–Terry loss and NLL loss, and proposes the LCPO method, which facilitates convergence in length control while reducing dependence on hyperparameters.
2. The method achieves strong results on both the 1.5B and 7B models across multiple benchmarks, reducing output length by over 50% while maintaining accuracy, and also demonstrates excellent performance on out-of-distribution (OOD) benchmarks (GPQA, MMLU).

**Weaknesses:**

1. In Section 3, the experiments on LRMs are conducted using a relatively “simple” dataset (L196), making the conclusions less representative. This is because, in LRMs, reflective behavior on more difficult problems should lead to both an increase in tokens and an improvement in accuracy. Moreover, the final training data only includes the “easy” subset, which may cause the model to overfit or “hack” on simple problems while losing its ability to allocate more tokens to harder ones.
2. There has been some research on difficulty-aware approaches, such as [1] AdaptThink and [2] Ada-R1, which reduces the novelty of this paper. Meanwhile, in RL-based methods, selecting the shortest response as the chosen one and the longest response as the rejected one is also a very common practice, as seen in works like [3] Kimi 1.5 and [4] Learning When to Think.
3.  Lacks evaluation on the AIME25 dataset, and AIME24 may have been exposed to the models used in this work.
4. For Table 2, I notice that the other methods use the different training time, could you provide the results in 50th step for other methods?

BTW, the anonymous code link provided in the paper is no longer accessible.

[1] AdaptThink: Reasoning Models Can Learn When to Think

[2] Ada-R1: Hybrid-CoT via Bi-Level Adaptive Reasoning Optimization

[3] Kimi k1.5: Scaling Reinforcement Learning with LLMs

[4] Learning When to Think: Shaping Adaptive Reasoning in R1-Style Models via Multi-Stage RL

**Questions:**

See the weaknesses.

---

> ### Author Response · Authors · 2025-11-23
> **Rebuttal by Authors (Part 1/4)**
>
> Dear Reviewer 4RRh,
>
> Thank you very much for your thorough review and constructive feedback! In our response, we have carefully addressed your comments with the following discussions.
> > *1. In Section 3, the experiments on LRMs are conducted using a relatively “simple” dataset (L196), making the conclusions less representative. This is because, in LRMs, reflective behavior on more difficult problems should lead to both an increase in tokens and an improvement in accuracy. Moreover, the final training data only includes the “easy” subset, which may cause the model to overfit or “hack” on simple problems while losing its ability to allocate more tokens to harder ones.*
>
> Thank you for the comment! We response to this point by point.
>
> **"In Section 3, the experiments on LRMs are conducted using a relatively “simple” dataset (L196), making the conclusions less representative. This is because, in LRMs, reflective behavior on more difficult problems should lead to both an increase in tokens and an improvement in accuracy" & "The final training data only includes the “easy” subset."**
>
> It appears that the term "simple" (and "easy") may have caused some misunderstanding, and we would like to clarify what we intended to convey.
>
> Here, "simple" refers to the fact that the model demonstrates strong reasoning performance on this **training set** (LIMR). According to the definition in Equation (2), this means the dataset is relatively "simple" specifically for the model being trained. In other words, the term reflects a **model-specific difficulty estimation**, rather than an objective measure of simplicity. As mentioned later in this section (line 215), the output length can be influenced by the model's perception of problem difficulty: it tends to generate more tokens when it perceives the reasoning as challenging. This is precisely the motivation behind our difficulty estimation and data filtering strategy. We aim to prevent the model from overly perceiving difficulty in problems it has already mastered (but objectively may not be easy). **From this perspective, our idea aligns with your point** that reflective behavior on more difficult problems should lead to both an increase in tokens and an improvement in accuracy in LRMs.
>
> In fact, the average generation lengths of the 7B and 1.5B models on this training set are 4,678 tokens and 7,030 tokens, respectively, which are close to their generation lengths on the Minerva-Math dataset (as shown in Table 3). However, both models achieve low accuracy below 40% on Minerva-Math. This indicates a contradiction between the model's perceived difficulty and its actual capability, leading it to generate excessively long responses even on problems that are not inherently difficult for itself (as there are short yet effective reasoning trajectories). Our goal is to correct the model's difficulty perception, thereby aligning its behavior with the desired "connection between difficulty levels and generation length". By doing so, we guide our model towards ***adaptive* shortening**.
>
> **"Moreover, the final training data only includes the “easy” subset, which may cause the model to overfit or “hack” on simple problems while losing its ability to allocate more tokens to harder ones."**
>
> We aim to prevent the model from overly perceiving difficulty in problems it has already mastered (labeled as "easy" but objectively may not be easy). Thus **the model is encouraged to avoid excessive generation on "simple" tasks only**.
>
> **Our method has several key features that help to avoid overfit**. From the data and ranking perspective (Section 3.2), our training relies on preference pairs constructed from the "easy" split, where all reasoning trajectories are correct. As a result, the contrast between preferred and rejected responses focuses specifically on the conciseness of reasoning patterns, rather than correctness. This allows the model to learn a more efficient reasoning *style*, rather than being constrained to a limited set of simple problem-solving *tricks* for problems it has already mastered. From the algorithmic perspective (Section 3.3.3), our method explicitly balances the implicit NLL reward, which helps prevent the model from collapsing toward the chosen responses. Furthermore, the use of small-scale data and training reduces the risk of overfitting.
>
> Our experimental results support our response:
>
> 1. Figure 4 (Section 4.3): After training, the model naturally uses more tokens for harder problems and fewer for easier ones, demonstrating adaptive reasoning behavior. **The actual generation length is aligned with difficulty annotation,** which further validates that our method resonates with your point on the connection between difficulty level and generation length.
>
> 2. Table 3: On challenging benchmarks like OlympiadBench, LCPO maintains accuracy while significantly reducing length. This indicates that the model does not lose its ability to reason deeply when necessary.

---

> ### Author Response · Authors · 2025-11-23
> **Rebuttal by Authors (Part 2/4)**
>
> 3. Table 5 (OOD generalization beyond the training distribution): On benchmarks that distinct from our training data, models trained with our method are still capable of reducing generation length by over 50%, and even even achieve a performance gain on GPQA-D.
>
> Thank you again for the valuable feedback! We have conducted additional experiments with a more challenging dataset, the amc-aime split from Eurus-2 dataset [0, 1]. Results on the new dataset are similar to that on LIMR (**Figure 6 in Appendix F, added to the revised PDF**). We will revise Section 3 to convey our idea more clearly.
> > *2. There has been some research on difficulty-aware approaches, such as [1] AdaptThink and [2] Ada-R1, which reduces the novelty of this paper. Meanwhile, in RL-based methods, selecting the shortest response as the chosen one and the longest response as the rejected one is also a very common practice, as seen in works like [3] Kimi 1.5 and [4] Learning When to Think.*
>
> Thank you for the comment and providing these valuable references!
>
> First of all, it should be noted that among the four works you mentioned, three of them are concurrent with our work, while the remaining one is a technical report in which the introduced techniques are not fully publicly available.
>
> We would like to clarify our main and unique contributions (summarized in Section 1), which we believe represent a novel and systematic approach to efficient reasoning. **Our work explores how to achieve efficient reasoning through small-scale training.** Under a unified Bradley-Terry loss framework, we provide theoretical insights into effectively aligning models with length preferences using limited data. With only limited data and training, our method achieves a reduction of over 50% in output length while maintaining reasoning performance. As shown in Table 1, these results differ significantly from those of prior works: 22k trajectories compared to strong RL baselines like L1 (645k) and TrEff (24.8k); $\sim$10.4 A100\*h for training the 7B model compared to $\sim$560 A100\*h of TrEff.
>
> "Difficulty-aware" is a very broad concept. While AdaptThink and Ada-R1 **aim to select between long or short CoT generation modes directly based on problem difficulty**, we aim to **reduce the generation length of LRMs with limited tuning**. Difficulty estimation in our method serves solely as a data annotation strategy. The model itself does not directly or explicitly assess the difficulty of a problem. **Our approach essentially corrects the misalignment between the model's intrinsic difficulty-adaptive behavior and the objective difficulty of problems**. They are only conceptually similar, but are entirely different.
>
> Kimi 1.5 and AutoThink ("Learning When to Think") do not directly select the shortest or longest answers for comparison. Instead, they **incorporate a regularization term calculated based on length into the reward function for online RL**. Our approach, on the other hand, aims to **maximize the implicit length preference signal during preference optimization**. While both aim to learn length preferences, they differ in methodology (online RL vs. preference optimization) and implementation.
>
> Moreover, difficulty estimation and preference ranking are just two module within our overall framework and do not undermine our core contribution.
>
> We fully acknowledge the effort and contributions made by previous studies. Thank you again for bringing these relevant works to our attention! Accordingly, we will properly cite these works in our subsequent revisions and supplement the related work section with discussions on the "hybrid CoT" lines of research such as Ada-R1. We further compared our method with AdaptThink and AutoThink, which have publicly available open-source weights. Results are as follows (other baselines from the paper are omitted for brevity and clarity):
>
> Cost of 7B model estimated in A100*h:
>
> |   Method   |          Type           | GPU hours* |
> | :--------: | :---------------------: | :--------: |
> | AdaptThink |           RL            |   ~1792    |
> | AutoThink  |           RL            |   ~5760    |
> |    Ours    | preference optimization |   ~10.4    |
>
> *GPU hours: Due to conversion discrepancies in GPU hours, the specific figures may fluctuate, but they can provide a reference in terms of magnitude.

---

> ### Author Response · Authors · 2025-11-23
> **Rebuttal by Authors (Part 3/4)**
>
> 1.5B model:
>
> |            |             math-500             |              gsm8k              |           minerva-math           |              aime24              |              amc23               |             olympiad             |                       |
> | :--------: | :------------------------------: | :-----------------------------: | :------------------------------: | :------------------------------: | :------------------------------: | :------------------------------: | :-------------------: |
> |  original  |          83.00  /  5665          |         86.28  /  2457          |          26.84  /  7211          |         29.17  /  16355          |         70.62  /  10162          |         44.66  /  11715          |           -           |
> | AdaptThink | 80.20 (-2.80)  /  1715 (-69.73%) | 82.64 (-3.64)  /  447 (-81.81%) | 25.00 (-1.84)  /  1910 (-73.51%) | 27.08 (-2.09)  /  8494 (-48.07%) | 68.28 (-2.34)  /  3188 (-68.63%) | 41.84 (-2.82)  /  4309 (-63.22%) |  -15.83  /  -67.50%   |
> | AutoThink  | 81.00 (-2.00)  /  1820 (-67.87%) | 79.98 (-6.30)  /  601 (-75.54%) | 27.57 (+0.73)  /  2397 (-66.76%) | 28.96 (-0.21)  /  7338 (-55.13%) | 67.34 (-3.28)  /  3487 (-65.69%) | 44.36 (-0.30)  /  4049 (-65.44%) |  -11.36  /  -66.07%   |
> |    Ours    | 83.20 (+0.20)  /  2397 (-57.69%) | 85.82 (-0.46)  /  946 (-61.50%) | 27.21 (+0.37)  /  2596 (-64.00%) | 29.58 (+0.41)  /  8810 (-46.13%) | 70.47 (-0.15)  /  4026 (-60.38%) | 44.81 (+0.15)  /  4921 (-57.99%) | **+0.52  /  -57.31%** |
>
> 7B model:
>
> |            |             math-500             |              gsm8k              |           minerva-math           |              aime24               |              amc23               |             olympiad             |                       |
> | :--------: | :------------------------------: | :-----------------------------: | :------------------------------: | :-------------------------------: | :------------------------------: | :------------------------------: | :-------------------: |
> |  original  |          92.20  /  4223          |         91.81  /  1677          |          36.76  /  5926          |          51.46  /  13411          |          87.97  /  6966          |          56.82  /  8789          |           -           |
> | AdaptThink | 90.60 (-1.60)  /  1975 (-53.23%) | 91.28 (-0.53)  /  354 (-78.89%) | 36.40 (-0.36)  /  2787 (-52.97%) | 53.54 (+2.08)  /  11002 (-17.96%) | 86.56 (-1.41)  /  4701 (-32.52%) | 55.93 (-0.89)  /  6594 (-24.97%) |   -2.71  /  -43.42%   |
> | AutoThink  | 89.40 (-2.80)  /  2187 (-48.21%) | 86.81 (-5.00)  /  809 (-51.76%) | 36.03 (-0.73)  /  2794 (-52.85%) | 50.42 (-1.04)  /  8215 (-38.74%)  | 83.91 (-4.06)  /  4005 (-42.51%) | 56.97 (-0.15)  /  4817 (-45.19%) |  -13.78  /  -46.54%   |
> |    Ours    | 91.40 (-0.80)  /  2033 (-51.86%) | 92.92 (+1.14)  /  796 (-52.53%) | 38.97 (+2.21)  /  2079 (-64.92%) | 48.75 (-2.71)  /  6892 (-48.61%)  | 86.88 (-1.09)  /  3108 (-55.38%) | 56.08 (-0.74)  /  4222 (-51.96%) | **-1.99  /  -54.21%** |
>
> Both methods serve as very strong baselines. Yet, our approach consistently demonstrates comparable performance under a lower cost.
>
> > *3. Lacks evaluation on the AIME25 dataset, and AIME24 may have been exposed to the models used in this work.*
>
> Thank you for the suggestion.
>
> We tested our method and all the baselines from Table 3 on this benchmark. Our method still works on AIME25. Our method reduces output length by 44% on the 7B model and by 56% on the 1.5B model, while the reasoning performance is preserved.
>
> 1.5B model:
>
> |   model    |                aime25                |
> | :--------: | :----------------------------------: |
> |   origin   |           22.71  /  15985            |
> |    CoD     |   22.08 (-0.63)  /  15313 (-4.20%)   |
> |  L1-exact  |  19.58 (-3.13%)  /  3092 (-80.66%)   |
> |   L1-max   |   21.46 (-1.25)  /  3481 (-78.22%)   |
> |   TrEff    |   22.50 (-0.21)  /  9807 (-38.65%)   |
> | AdaptThink |   22.92 (+0.21)  /  8697 (-45.59%)   |
> | AutoThink  |   21.67 (-1.04)  /  7593 (-52.50%)   |
> |    LCPO    | **22.23 (-0.48)  /  7194 (-56.00%)** |
>
> 7B model:
>
> |   model    |                aime25                 |
> | :--------: | :-----------------------------------: |
> |   origin   |            37.50  /  14622            |
> |    CoD     |   36.25 (-1.25)  /  13069 (-10.62)    |
> |  L1-exact  |   27.29 (-10.21)  /  4075 (-72.13%)   |
> |   L1-max   |   28.75 (-8.75)  /  3063 (-79.05%)    |
> |   TrEff    |   33.75 (-3.75)  /  11156 (-23.70%)   |
> |    DAST    |   37.29 (-0.21)  /  15255 (+4.33%)    |
> | AdaptThink |  37.08 (-0.42%)  /  12319 (-15.75%)   |
> | AutoThink  |   36.67 (-0.83)  /  8913 (-39.04%)    |
> |    LCPO    | **36.87 (-0.63)  /   8187 (-44.01%)** |

---

> ### Author Response · Authors · 2025-11-23
> **Rebuttal by Authors (Part 4/4) and References**
>
> > *4. For Table 2, I notice that the other methods use the different training time, could you provide the results in 50th step for other methods?*
>
> Thank you for your careful and thorough review. We sincerely apologize for this oversight. It was a typo introduced during our internal revision process. The results presented in Table 2 are indeed based on models at the 50th training step. We will fix it in the revision.
>
>
>
> > *5. The anonymous code link provided in the paper is no longer accessible.*
>
> Thank you for your careful check. Our anonymous code link remains valid. However, due to LaTeX formatting, a backslash was added before the underscore in the URL. This may cause some browsers to misinterpret the path. To access the link correctly, please simply remove any backslashes before the underscores. For your convenience, we are providing the correct link again below:
>
>  https://anonymous.4open.science/r/anonymous_code_74CD
>
>
>
> References:
>
> [0] Cui, Ganqu, et al. "Process reinforcement through implicit rewards." *arXiv preprint arXiv:2502.01456* (2025).
>
> [1] Yuan, Lifan, et al. "Free process rewards without process labels." *arXiv preprint arXiv:2412.01981* (2024).

---

> ### Author Response · Authors · 2025-11-28
>
> Dear Reviewer,
>
> We sincerely appreciate your time and effort in reviewing our work and providing valuable feedback. Your insights have been instrumental in helping us improve the manuscript.
>
> In response to your comments, we have made our best efforts to address each point comprehensively.
> We have also revised our manuscript based on the rebuttal. The specific modifications we made in the revised PDF are as follows:
>
> **1. Revised potentially confusing wording in Section 3.2;**
>
> **2. Fixed a typo in Appendix B;**
>
> **3. Introduced a replication of the experiments from Section 3.2 on a more challenging dataset in Appendix F.1;**
>
> **4. Introduced related work introduction along with corresponding citations and experimental comparison with Hybrid-CoT in Appendix H.**
>
> As the author-reviewer discussion period is drawing to a close, we kindly wish to confirm whether there remain any unresolved concerns or points requiring further clarification. We would be delighted to engage in deeper discussions to ensure all aspects are thoroughly addressed.
>
> Thank you again for your thoughtful review. We greatly value your expertise and look forward to your feedback.
>
> Best regards,
>
> The Authors

---

### Official Review · Reviewer_HaQu · 2025-11-02

**Soundness:** 3
**Presentation:** 3
**Contribution:** 3
**Rating:** 6
**Confidence:** 3

**Summary:**

This paper addresses the “overthinking” problem in large reasoning models (LRMs) such as DeepSeek-R1, which often produce excessively long reasoning chains. The authors propose Length Controlled Preference Optimization (LCPO) — a small-scale, offline tuning method under a unified Bradley–Terry framework — that explicitly counterbalances the implicit NLL reward to align model preferences toward concise reasoning trajectories. Empirical results on multiple math reasoning benchmarks (MATH-500, GSM8K, Minerva-Math, AIME24, AMC23, OlympiadBench) show over 50% reduction in output length with minimal accuracy drop and good out-of-domain generalization to MMLU and GPQA-D.

**Strengths:**

1. The issue of long, inefficient chain-of-thought generation is increasingly important for practical deployment of reasoning models.

2. The use of a Bradley–Terry loss analysis to interpret and modify preference optimization objectives (e.g., DPO, ORPO, SimPER) is clear and insightful.

3. LCPO achieves competitive results with only ~0.8K training samples and 50 steps — a strong practical contribution.

4. Strong empirical validation: Evaluations across multiple datasets (both in-domain and OOD) support the claimed efficiency and generalizability.

5. Clear ablation design: The authors systematically analyze the impact of data difficulty filtering and algorithmic components.

**Weaknesses:**

1. Does LCPO risk collapsing reasoning diversity — e.g., over-shortening reasoning even for difficult problems? It would be interesting to measure the diversity of generations.

2. OOD evidence is limited to MMLU and GPQA-D, without baseline comparisons, making the generalization beyond math unclear.

3. The paper emphasizes using only a small number of preference samples, but the actual workflow involves a significant upfront computational cost: it requires multiple rollouts across the dataset. In other words, while the training phase uses limited data, generating this data itself demands substantial computational resources.

**Questions:**

see above weaknesses

---

> ### Author Response · Authors · 2025-11-23
> **Rebuttal by Authors (Part 1/2)**
>
> Dear Reviewer HaQu,
>
> Thank you very much for your thorough review and constructive feedback! In our response, we have carefully addressed your comments with the following discussions.
> > *1. Does LCPO risk collapsing reasoning diversity — e.g., over-shortening reasoning even for difficult problems? It would be interesting to measure the diversity of generations.*
>
> Thank you for your insightful feedback and suggestion!
>
> We provide arguments below to show that our method **achieves *adaptive* shortening** rather than a collapse of reasoning diversity. Furthermore, we follow your excellent suggestion to directly measure the diversity of  generations and present the results.
>
> Our data construction pipeline (Section 3.2, line 212-229) is designed to prevent diversity collapse. By filtering data based on estimated difficulty (easy/medium/hard), we are not enforcing a universal shortness but rather teaching the model that for a given "easy" (easy enough for the model itself) problem, a shorter correct path exists. The model generalizes this principle, learning to find efficient paths across the difficulty spectrum without being forced to use the same path for all problems.
>
> Our results in Section 4.3 and Figure 4 directly address this concern. The data shows that our method reduces generation length proportionally to problem difficulty. **After training, the average generation length for the most difficult problems (Level 5, the highest difficulty level) remains significantly longer than for easier problems (Level 1-3).** This demonstrates that the model has not collapsed into a one-size-fits-all short response but has learned to allocate less reasoning effort to easier problems while reserving sufficient effort for difficult ones. The maintenance of high accuracy on notoriously difficult benchmarks like OlympiadBench (Table 3) also serves as an indirect evidence that reasoning diversity for hard problems is preserved.
>
> To quantitatively measure the diversity of generations, we adopted the following metrics:
>
> Distinct-n [0, 1]: a metric used to calculate the diversity of a group of sentences,
>
> Expectation-Adjusted-Distinct (EAD) [2]: an improved Distinct metric that removes the biases of the original Distinct score (Distinct tends to assign lower scores to longer sequences),
>
> and conducted additional experiments on MATH-500, which was chosen for its diverse and annotated difficulty levels. We generated 16 responses for each problem using the same generation settings as Table 3. Our results are as follows:
>
> |                         | Distinct-1 $\uparrow$ | Distinct-2 $\uparrow$ | Distinct-3 $\uparrow$ | EAD $\uparrow$ |
> | :---------------------: | :-------------------: | :-------------------: | :-------------------: | :------------: |
> |       Original-7B       |        0.0222         |        0.1132         |        0.2320         |     0.0256     |
> |         Ours-7B         |      **0.0292**       |      **0.1260**       |      **0.2375**       |   **0.0312**   |
> |      Original-1.5B      |        0.0199         |        0.1036         |        0.2179         |     0.0240     |
> |        Ours-1.5B        |      **0.0285**       |      **0.1316**       |      **0.2553**       |   **0.0313**   |
> |  Original-7B on level5  |        0.0046         |        0.0252         |        0.0546         |     0.0058     |
> |    Ours-7B on level5    |      **0.0062**       |      **0.0303**       |      **0.0613**       |   **0.0070**   |
> | Original-1.5B on level5 |        0.0040         |        0.0222         |        0.0498         |     0.0055     |
> |   Ours-1.5B on level5   |      **0.0063**       |      **0.0322**       |      **0.0668**       |   **0.0074**   |
>
> As shown in the table above, **the model trained with our method achieves a higher score under all four metrics**. This indicates that our method has improved generation diversity while simultaneously reducing overall response length.
>
> > *2. OOD evidence is limited to MMLU and GPQA-D, without baseline comparisons, making the generalization beyond math unclear.*
>
> We thank you for the insightful comment! We have conducted additional experiments to demonstrate the broad applicability of our method.
>
> The MMLU test set spans subjects in the humanities, social sciences, hard sciences, and many other areas beyond mathematics. The GPQA-Diamond (GPQA-D) test set comprises expert-level questions in biology, physics, and chemistry. The broad competencies covered by these two datasets, extending beyond mathematical skills, already provide strong evidence of  generalization to out-of-domain (OOD) scenarios in our view.
>
> Nevertheless, we found your suggestion highly valuable and have therefore added one more OOD benchmark:
>
> Winogrande [3]: a collection of fill-in-a-blank commonsense reasoning problems that is widely used to evaluate models' reasoning capabilities in general scenarios [4, 5, 6]

---

> ### Author Response · Authors · 2025-11-23
> **Rebuttal by Authors (Part 2/2)**
>
> and tested all the baselines from Table 3 (main results) under the same settings. Results are as follows:
>
> 1.5B model:
>
> |  model   |                MMLU                 |                GPQA-D                |             Winogrande              |                       |
> | :------: | :---------------------------------: | :----------------------------------: | :---------------------------------: | :-------------------: |
> | original |           46.43  /  2549            |           34.85  /  10312            |           50.04  /  1516            |           -           |
> |   CoD    |  46.72 (+0.29)  /  1592 (-37.54%)   |   33.33 (-1.52)  /  7328 (-28.94%)   |   46.88 (-3.16)  /  1572 (+3.69%)   |   -4.39  /  -20.93%   |
> | L1-exact |  48.64 (+2.21)  /  5373 (+52.56%)   |   31.31 (-3.54)  /  3253 (-68.45%)   |  51.14 (+1.10)  /  4819 (+217.88%)  |   -0.23  /  +67.33%   |
> |  L1-max  |  49.29 (+2.86)  /  2219 (-12.95%)   |   31.82 (-3.03)  /  2839 (-72.47%)   |  50.59 (+0.55)  /  2409 (+58.91%)   |   +0.38  /  -8.84%    |
> |  TrEff   |   44.80 (-1.63)  /  1201(-52.88%)   |   27.27 (-7.58)  /  6914 (-32.95%)   |   49.09 (-0.92)  /  795 (-47.56%)   |  -10.13  /  -44.46%   |
> |   LCPO   | **46.78 (+0.35)  /  760 (-70.18%)** | **39.39 (+4.54)  /  4340 (-57.91%)** | **50.83 (+0.79)  /  309 (-79.62%)** | **+5.68  /  -69.24%** |
>
> 7B model:
>
> |  model   |                MMLU                 |                GPQA-D                |             Winogrande              |                       |
> | :------: | :---------------------------------: | :----------------------------------: | :---------------------------------: | :-------------------: |
> | original |           65.27  /  1858            |            48.99  /  9237            |           61.17  /  1042            |           -           |
> |   CoD    |   63.72 (-1.55)  /  822 (-55.76%)   |   47.98 (-1.01)  /  5834 (-36.84%)   |   62.12 (+0.95)  /  505 (-51.54%)   |   -1.61  /  -48.05%   |
> | L1-exact |  63.77 (-1.50)  /  2834 (+52.53%)   |   46.46 (-2.53)  /  3235 (-64.98%)   |  62.90 (+1.73)  /  2949 (+64.67%)   |   -2.30  /  +17.41%   |
> |  L1-max  |   64.02 (-1.25)  /  983 (-47.09%)   |   46.97 (-2.02)  /  2039 (-77.93%)   |   62.19 (+1.02)  /  847 (-18.71%)   |   -2.25  /  -47.91%   |
> |  TrEff   |  63.17 (-2.10)  /  1077 (-42.03%)   |   44.95 (-4.04)  /  6470 (-29.96%)   |   60.22 (-0.95)  /  472 (-54.70%)   |   -7.09  /  -42.43%   |
> |   DAST   |  65.62 (+0.35)  /  2468 (+32.83%)   |  47.98 (-1.01)  /  10819 (+17.13%)   |  59.35 (-1.82)  /  1630 (+56.43%)   |   -2.48  /  +35.46%   |
> |   LCPO   | **64.16 (-1.11)  /  835 (-55.06%)** | **52.53 (+3.54)  /  4301 (-53.44%)** | **62.12 (+0.95)  /  375 (-64.01%)** | **+3.38  /  -57.50%** |
>
> Our method maintains strong generalization in out-of-distribution scenarios and achieves the best trade-off between reasoning capability and generation length.
>
> We will add these results to Section 4.3 (Generalizability to OOD Scenarios) in the revised manuscript.
>
> > *3. The paper emphasizes using only a small number of preference samples, but the actual workflow involves a significant upfront computational cost: it requires multiple rollouts across the dataset. In other words, while the training phase uses limited data, generating this data itself demands substantial computational resources.*
>
> Thank you for this insightful observation!
>
> We agree that the rollout process is the primary computational cost of our method. However, we would like to clarify and contextualize this cost:
>
> 1. The rollout is a one-time, offline process. Once the preference dataset is generated, it can be reused for multiple training runs and easily shared (e.g., via platforms like HuggingFace).
> 2. Compare to offline methods (in Table1), we need less data from rollout. Compare to online RL methods  (also in Table1), which necessitate a continuous and interleaved rollout and policy updating throughout the entire training process, the overall computational budget (rollout + training) of our method remains highly competitive, especially considering the performance we achieved.
>
> We provide some intuitive numerical comparisons:
>
> 1. As detailed in Table 1 (column "Data"), our method requires less data from rollout generation (22k trajectories) compared to strong baselines like L1 (645k) and TrEff (24.8k).
> 2. For the 7B model training, TrEff spent $\sim$560 A100\*h for training the 7B model, while our method spent $\sim$10.4 A100\*h in total. Furthermore, since the dataset is reusable, no additional time is consumed for data rollout during repeated experiments.
>
> A core contribution of our work is the demonstration that effective length compression can be achieved with small-scale preference optimization, provided a diverse set of self-generated trajectories is available. This finding is independent of the initial data generation cost.

---

> ### Author Response · Authors · 2025-11-23
> **References**
>
> References:
>
> [0] Li, Jiwei, et al. "A diversity-promoting objective function for neural conversation models." *Proceedings of the 2016 conference of the North American chapter of the association for computational linguistics: human language technologies*. 2016.
>
> [1] Wang, Chaoqi, et al. "Beyond Reverse KL: Generalizing Direct Preference Optimization with Diverse Divergence Constraints." *The Twelfth International Conference on Learning Representations*.
>
> [2] Liu, Siyang, et al. "Rethinking and Refining the Distinct Metric." *Proceedings of the 60th Annual Meeting of the Association for Computational Linguistics (Volume 2: Short Papers)*. 2022.
>
> [3] Sakaguchi, Keisuke, et al. "Winogrande: An adversarial winograd schema challenge at scale." *Proceedings of the AAAI Conference on Artificial Intelligence*. Vol. 34. No. 05. 2020.
>
> [4] Grattafiori, Aaron, et al. "The llama 3 herd of models." *arXiv preprint arXiv:2407.21783* (2024).
>
> [5] Team, Gemma, et al. "Gemma 3 technical report." *arXiv preprint arXiv:2503.19786* (2025).
>
> [6] Team, Kimi, et al. "Kimi k2: Open agentic intelligence." *arXiv preprint arXiv:2507.20534* (2025).

---

> ### Author Response · Authors · 2025-11-28
>
> Dear Reviewer,
>
> We sincerely appreciate your time and effort in reviewing our work and providing valuable feedback. Your insights have been instrumental in helping us improve the manuscript.
>
> In response to your comments, we have made our best efforts to address each point comprehensively. We have also revised our manuscript based on the rebuttal. The specific modifications we made in the revised PDF are as follows:
>
> **1. Adjusted the description of experimental content in Section 4.3;**
>
> **2. Added a new OOD dataset Winogrande in Appendix A;**
>
> **3. Introduced OOD generalization experiments with expanded datasets and baselines in Appendix G;**
>
> **4. Introduced experiments exploring the impact of LCPO on generation diversity in Appendix I.**
>
> As the author-reviewer discussion period is drawing to a close, we kindly wish to confirm whether there remain any unresolved concerns or points requiring further clarification. We would be delighted to engage in deeper discussions to ensure all aspects are thoroughly addressed.
>
> Thank you again for your thoughtful review. We greatly value your expertise and look forward to your feedback.
>
> Best regards,
>
> The Authors

---

### Author Response · Authors · 2025-11-27
**Reminders about Revision**

We have revised our manuscript according to the rebuttal and uploaded the revised PDF. The modifications have been highlighted in **blue**. Please note that the main content of the paper remains unchanged; the revisions we made primarily consist of wording adjustments and the addition of supplementary experiments from the rebuttal to the appendix.

The specific modifications we made are as follows:

1. Revised potentially confusing wording in Section 3.2 (Reviewer 4RRh);

2. Clarified the insensitivity to hyperparameter $\lambda$ in Section 3.3.3 (Reviewer jTBW);

3. Adjusted the description of experimental content in Section 4.3 (Reviewer HaQu);

4. Added the new dataset Winogrande in Appendix A (Reviewer HaQu);

5. Fixed a typo in Appendix B (Reviewer 4RRh);

6. Introduced a replication of the experiments from Section 3.2 on a more challenging dataset in Appendix F.1 (Reviewer 4RRh);

7. Introduced hyperparameter sensitivity analysis experiments for LCPO and comparative experiments between LCPO and ORPO in Appendix F.2 (Reviewer jTBW);

8. Introduced OOD generalization experiments with expanded datasets and baselines in Appendix G (Reviewer HaQu);

9. Introduced related work introduction and experimental comparison with Hybrid-CoT in Appendix H (Reviewer 4RRh);

10. Introduced experiments exploring the impact of LCPO on generation diversity in Appendix I (Reviewer HaQu);

***

Dear reviewers,

We respect all reviewers and the area chair, and thank you for the time and effort you have dedicated to organizing and reviewing. : )

Out of respect for the reviewers, we have made our best efforts to provide comprehensive responses to address the concerns. For open-ended questions, we have also provided thorough responses based on our technical understanding and survey conducted promptly during the rebuttal period. We eagerly look forward to further in-depth discussions with you! : )

Sincerely,

The Authors

---

### Author Response · Authors · 2025-12-01
**Summary of Rebuttal by Authors (Part 1/3)**

Dear Program Chairs, Senior Area Chairs, Area Chairs and Reviewers,

We sincerely thank you for your thorough review and constructive feedback! During the rebuttal period, we conduct extensive additional analyses and experiments to address the reviewers' concerns and questions. Below is a summary of our major response efforts.

***

We sincerely thank the reviewers for their insightful and positive feedback, and their time during the review process! We are excited that the reviewers recognized the strengths of our paper:

1.  **Effective and Practical Length Reduction:** Our proposed method successfully reduces reasoning length by over 50% while maintaining accuracy, addressing the "overthinking" problem with high training efficiency (requiring only ~0.8K samples and 50 steps). (HaQu, 4RRh, jTBW, og6S)

2.  **Strong and Generalizable Empirical Performance:** The method demonstrates competitive or strong results on multiple in-domain and out-of-distribution (OOD) benchmarks (e.g., MMLU, GPQA), proving its robustness and generalizability. (HaQu, 4RRh, jTBW)

3.  **Novel and Insightful Methodological Foundation:** The paper provides an insightful theoretical analysis under a unified Bradley-Terry loss framework from an empirical perspective. We reframing length control by mining the model's own short reasoning paths and using a Bradley-Terry based loss. (HaQu, 4RRh, og6S, jTBW)

4.  **Extensive Experiments, Clear and Comprehensive Experimental Design:** The work includes extensive baselines (SFT, DPO variants), clear ablations studies on data filtering and algorithmic components, and thorough OOD tests, providing strong validation. (HaQu, jTBW)

---

> ### Author Response · Authors · 2025-12-01
> **Summary of Rebuttal by Authors (Part 2/3)**
>
> We also thank the reviewers for their insightful raised concerns. We address these concerns as follows:
>
> ***
>
> ### Summary for Reviewer HaQu:
>
> **W1: Potential Collapse of Reasoning Diversity**
>
> We demonstrate that our method achieves difficulty-adaptive shortening through the inherent characteristics of the method, the experiments in Figure 4 of Section 4.3, and by directly measuring the generative diversity using Distinct-n and EAD.
>
> **W2: OOD Generalization Evidence**
>
> We substantially expand OOD evaluation by adding Winogrande and providing comprehensive baseline comparisons across three diverse benchmarks. Results confirm our method maintains strong generalization beyond mathematics while achieving the best capability-efficiency trade-off.
>
> **W3: Upfront Computational Cost**
>
> We clarify that rollout is a one-time, offline process producing reusable data, with less rollout data than key baselines. Also, the  overall computational budget remains substantially lower. To illustrate these points, we provide intuitive numerical comparisons.
>
> ***
>
> ### Summary for Reviewer 4RRh:
>
> **W1.1: Inappropriate Dataset for Experiment in Section 3.2**
>
> We clarify that the dataset LIMR is a training set. And the word "simple" refers to model-specific difficulty from the perspective of its actual capability rather than objective simplicity. Our method precisely addresses the misalignment between the model's perceived difficulty and its actual capability on simple problems. Also, we re-run the same experiment on a more challenging dataset (AMC-AIME split from Eurus-2) to demonstrate that the conclusion still holds.
>
> **W1.2: Potential Overfit on Simple Problems**
>
> We demonstrate that training on the "easy" split focuses on learning concise reasoning patterns rather than easy-problem-only tricks. Results of experiments shown in Figure 4, Section 4.3, results on challenging benchmarks (e.g. Olympiad) in Table 3 and OOD results support our point.
>
> **W2: Novelty Compared to Existing Works**
>
> While the four mentioned papers (a technical report and three concurrent works) **differ significantly** from our approach, we have respectfully explored potential commonalities and elaborated on the key distinctions in our rebuttal. Furthermore, **we have properly cited these papers and have included additional related work discussions along with extensive experimental comparisons (cost and capability-efficiency) in Appendix H**. To comply with the strict page limits, we could not place this in the main text, but we would be happy to incorporate it should space permit in the future.
>
> **W3: AIME25 Dataset**
>
> We add additional experiments (with all the baselines including those in W2) on AIME25. Our method remains highly effective and competitive.
>
> **W4: Results of 50th Checkpoints in Table 2**
>
> We fix the typo of the training step (350th $\rightarrow$ 50th).
>
> ***
>
> ### Summary for Reviewer og6S:
>
> **W1: Generalization of ID Preference Alignment**
>
> We clarify that leveraging a model's intrinsic capabilities is a fair, necessary, and effective setup that aligns with RL **in reasoning tasks, especially challenging ones**, rather than a drawback.
>
> **W2: Stability of LCPO**
>
> We clarify that our method achieves competitive performance on reasoning tasks with only small-scale training, unlike other approaches that require large volumes of data. While training on larger datasets is not necessary, our method can easily adapt to such scenarios through straightforward hyperparameter adjustments when needed.
>
> **W3: Lack comparison with RL-based and prompt-based methods.**
>
> We clarify that actually we have already included these two types of method for comparison (listed in Table 1, experiments results in Table 3 of the original paper and Table 10 of the revised paper).

---

> ### Author Response · Authors · 2025-12-01
> **Summary of Rebuttal by Authors (Part 3/3)**
>
> ### Summary for Reviewer jTBW:
>
> **W1 & Q1: Algorithmic Novelty**
>
> We clarify the algorithm is a part of our propose pipeline. And we clarify the differences between LCPO and ORPO:
>
> 1. LCPO is a theoretically grounded, simplified objective for efficient reasoning, unlike ORPO which intuitively adds a penalty to SFT for general instruction-following tasks;
> 2. While LCPO removes the NLL loss (SFT) as it can hinder length preference alignment in reasoning, ORPO retains it.
>
> We also add the suggested comparison experiment where the hyperparameter $\lambda$ is scaling up. Results demonstrate the advantage of LCPO:
>
> 1. **Hyperparameter Robustness**: LCPO performs consistently well without tuning $\lambda$; ORPO is sensitive and requires careful $\lambda$ search and selection.
> 2. **Performance**: LCPO outperforms ORPO, especially as $\lambda$ increases, avoiding overfitting and maintaining accuracy.
> 3. **Diversity**: LCPO improves generation diversity (measured by Distinct-n and EAD) while shortening responses, whereas ORPO’s diversity falls behand LCPO with high $\lambda$.
>
> **W2 & Q2: Data Efficiency**
>
> We clarify that the number of rollouts is a standard setting in related work and is flexible based on budget. Our rollout is a one-time, offline cost, with superior overall efficiency.
>
> Filtering synthetic data for down-streaming purposes is a common practice in reasoning tasks. We keep only the easy split to prevent the model from over-generating on mastered problems.
>
> Answers to questions: For 7B model, 43.7% of rollout time is spent on medium/difficult questions. A two-stage proxy is not currently reliable. And performance prediction still requires model generation.
>
> **W3: Sensitivity Analysis**
>
> We add an experiments for sensitivity analysis. Results demonstrated that LCPO performs consistently well without tuning $\lambda$. Its actually a hyperparameter-free objective for efficient reasoning.
>
> **Q3: Generalization of ID Preference Alignment**
>
> We clarify that leveraging a model's intrinsic capabilities is a fair, necessary, and effective setup that aligns with RL **in reasoning tasks, especially challenging ones**, rather than a drawback. Unlike general scenarios, due to the lack of an ace model in efficient reasoning (shorter and stronger in challenging reasoning tasks), distillation from another model is unreliable.
>
> ***
>
> In response to the reviews and to provide further clarification, we have made the following revisions to our paper. Revisions are highlighted in blue for clarity:
>
> 1. **Revised potentially confusing wording in Section 3.2** (Reviewer 4RRh);
> 2. **Clarified the insensitivity to hyperparameter λ in Section 3.3.3** (Reviewer jTBW);
> 3. **Adjusted the description of experimental content in Section 4.3** (Reviewer HaQu);
> 4. **Added the new dataset WinoGrande in Appendix A** (Reviewer HaQu);
> 5. **Fixed a typo in Appendix B** (Reviewer 4RRh);
> 6. **Introduced a replication of the experiments from Section 3.2 on a more challenging dataset in Appendix F.1** (Reviewer 4RRh);
> 7. **Introduced hyperparameter sensitivity analysis experiments for LCPO and comparative experiments between LCPO and ORPO in Appendix F.2** (Reviewer jTBW);
> 8. **Introduced OOD generalization experiments with expanded datasets and baselines in Appendix G** (Reviewer HaQu);
> 9. **While the four cited papers differ significantly from our approach, we introduced related work introduction and experimental comparison with Hybrid-CoT in Appendix H** (Reviewer 4RRh);
> 10. **Introduced experiments exploring the impact of LCPO on generation diversity in Appendix I** (Reviewer HaQu);
>
> ***
>
> Out of respect for the reviewers, we have made our best efforts to provide comprehensive responses to address the concerns. For open-ended questions, we have also provided thorough responses based on our technical understanding and survey conducted promptly during the rebuttal period.
>
> We are genuinely excited and actively look forward to further in-depth discussions with all reviewers. We eagerly hope to receive additional feedback from the reviewers. Unfortunately, before we could receive comments from every reviewer, a serious incident on OpenReview prevented reviewers from providing further feedback.
>
> We sincerely hope that the Area Chair will take our rebuttal into full consideration. We also extend our heartfelt understanding and respect to the Area Chair for undertaking such significant responsibilities during this exceptional period.
>
> Thanks again for the effort and time dedicated to organizing and reviewing!

---

### Meta-Review · Area_Chair_9Ffp · 2026-01-05

**Summary:**

This paper presents an efficient method for efficient reasoning that achieves strong performance using only small-scale preference data. The submission was evaluated by four domain experts, who raised concerns regarding the method’s generalization, novelty, and empirical effectiveness. The AC has reviewed the authors’ rebuttal and finds that most of these concerns have been adequately addressed.

However, the AC thinks that this paper should be further strengthened with more experimental results. Specifically, results on AIME 2025 and HLE Math should be included in the main paper, not just AIME 2024, to better demonstrate temporal robustness and broader applicability. Besides, models larger than 7B should be evaluated to show the efficacy on stronger baselines, such as the latest Qwen3 series and QwQ-32B. Additionally, the rebuttal does not address coding performance, a key aspect of generalization. The authors are thus suggested to include the results of LiveCodeBench (or a comparable benchmark) to assess this point.

To maximize impact and reproducibility, the AC also recommends that the authors open-source their implementation (the provided link has expired). Provided these revisions are made and the limited evaluation scope, the AC moderately recommends Accept, but wouldn't mind if the paper gets rejected.

**Reviewer Concerns:**

1. Potential Collapse of Reasoning Diversity:  The authors have demonstrated that the method achieves difficulty-adaptive shortening through the inherent characteristics of the method.

2. OOD Generalization: Three extra benchmarks have been compared in the rebuttal. However, the AC thinks that it would be better if coding scenarios could be considered, such as the livecodebench.

3. Computational Cost: The authors have shown the efficacy of the proposed method in the rebuttal.

4. Inappropriate Dataset: While some experiments have been conducted in the rebuttal, such as the AMC-AIME split from Eurus-2, the AC encourages the authors to supplement harder datasets, such as AIME 25 and HLE-Math, into the final version.

5. Potential Overfit: The results on challenging benchmarks (e.g., Olympiad) in Table 3 and OOD results show the robustness of the method.

6. Novelty issue: The AC has checked the rebuttal and found that the authors have made necessary discussions regarding the references mentioned by the reviewer, and believes that the contribution of this paper is sufficient.

7. Stability: The authors have supplemented the results obtained from only small-scale training and shown that the proposed method can be scaled to larger training data.

8. Comparison with RL-based and prompt-based methods: The authors have clarified the results are already in the paper.

9. Clarity issues: The authors have shown different baselines (SFT, DPO variants), and ablation studies on data filtering and algorithmic components, and thorough OOD tests, and promised to fix the written issues.

**Reviewer Scores:**

The initial scores are 6/4/6/4, with corresponding confidence levels of 3/4/4/3.

After reviewing the authors’ rebuttal, the AC finds that the primary concern raised by reviewers has been largely addressed. In light of this, the AC believes the existing scores of 6 remain, and one of the 4s may be upgraded to 6s, resulting in a moderately positive assessment of the submission.

---

### Decision · Program_Chairs · 2026-01-26

Accept (Poster)